# *RAYQUAZA*[*]: Input-Conditioned Radial Basis Decomposition for Efficient Time-Series Forecasting

## Abstract

Time-series forecasting presents a persistent trade-off between simple, scalable linear models that struggle with complex dynamics and large neural architectures that offer high accuracy at a steep computational cost. We introduce **RAYQUAZA**, a parameter-efficient architecture designed to fill this gap. **RAYQUAZA** learns an adaptive basis decomposition of the signal into three complementary components: a smooth trend extractor, a residual correction branch, and a novel input-conditioned radial basis function layer. The iRBF module dynamically learns a compact set of localized Gaussian atoms for each input sequence, enabling it to model transient, non-stationary patterns like structural breaks and spikes that challenge simpler methods. With fewer than 0.12M parameters, **RAYQUAZA** achieves state-of-the-art accuracy on large-scale public benchmarks and demonstrates consistently strong performance across a diverse range of forecasting domains. Crucially, it outperforms lightweight linear baselines in the majority of long-horizon forecasting scenarios while remaining two to three orders of magnitude smaller than transformer-based models. These results establish **RAYQUAZA** as a practical, interpretable, and efficient model, proving that adaptive basis representations can deliver high accuracy without sacrificing efficiency

## 1 Introduction

Learning effective representations of sequential data is a foundational challenge in machine learning. For time-series, this manifests as a persistent trade-off. On one side are classical methods, such as the ARIMA family of models proposed by Box et al. (2015), which rely on rigid, interpretable representations like polynomial trends or fixed seasonalities. While scalable, these approaches struggle to capture the complex dynamics, non-stationarity, and abrupt structural changes common in real-world data. On the other side are large neural architectures. Recent transformer-based models from Zhou et al. (2021), Zhou et al. (2022b), and Nie et al. (2023a) learn powerful, high-dimensional latent representations and have achieved state-of-the-art accuracy. However, this performance comes at a high computational cost, posing significant barriers to deployment in resource-constrained settings. This gap motivates the need for approaches that are both expressive and efficient.

To bridge this gap, we introduce **RAYQUAZA**, a parameter-efficient architecture that learns an adaptive basis decomposition of the input signal. Unlike methods that rely on fixed bases, such as the N-BEATS model by Oreshkin et al. (2019a), or on uninterpretable latent spaces, **RAYQUAZA** decomposes the signal into three complementary components: a smooth trend extractor, a residual correction branch, and, at its core, a novel input-conditioned radial basis function (iRBF) layer.

The core innovation lies in the iRBF module, which dynamically learns a compact set of localized Gaussian atoms whose parameter centers, widths, and amplitudes are conditioned on each input sequence. Effectively, the module employs a small MLP as a hypernetwork that generates a specialized basis for each input, yielding a parsimonious and highly adaptive representation of the signal's transient components. This design, inspired by the universal approximation capabilities of RBF networks demonstrated by Park & Sandberg (1991) and Poggio & Girosi (1990), goes beyond the global fixed bases used in prior work. For instance, while BasisFormer by Ni et al. (2024) attends

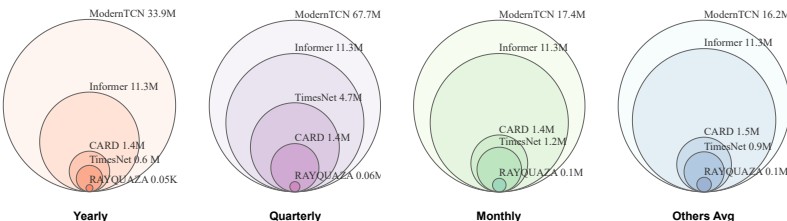

Figure 1: Model parameter counts across different forecasting granularities (Yearly, Quarterly, Monthly, Others) on the M4 dataset. **RAYQUAZA** consistently achieves the lowest parameter count, demonstrating superior parameter efficiency.

over a global basis set, **RAYQUAZA** allows the basis functions themselves to be a function of the input.

Through this *per-instance basis adaptation*, the iRBF module provides **RAYQUAZA** the flexibility to model both global structures and localized variations such as spikes, seasonal drifts, and structural breaks without requiring deep architectures or large parameter budgets. Moreover, the iRBF module itself is inherently interpretable: By visualizing the learned Gaussian atoms, one can directly inspect which temporal structures the model uses to construct its forecast, offering a level of transparency rarely seen in high-performing neural models.

We validate the effectiveness of our approach through a comprehensive and rigorous evaluation on three distinct classes of public benchmarks: the classic M4 dataset (100,000 series), the large-scale TFB benchmark (3,428 series), and the four widely used ETT datasets for long-horizon forecasting. Our contributions are as follows.

- We propose **RAYQUAZA**, an architecture that learns an adaptive representation of time-series by decomposing signals into a set of input-conditioned basis functions.
- We introduce a novel input-conditioned RBF (iRBF) layer that dynamically generates a compact set of Gaussian atoms, enabling both fine-grained modeling and interpretable inspection of learned temporal patterns.
- We demonstrate that this representation is highly effective, showing that **RAYQUAZA** achieves state-of-the-art accuracy on the large-scale TFB benchmark against a wide spectrum of prominent models.
- We further validate **RAYQUAZA**'s expressiveness in a rigorous channel-wise evaluation. The results confirm **RAYQUAZA**'s ability to crucially outperform lightweight linear baselines on complex, long-horizon tasks where they typically fail.

## 2 RELATED WORK

Time-series forecasting spans a wide spectrum of techniques, from classical statistical models to deep neural architectures and transformer-based designs. Although many recent methods achieve strong accuracy, they often do so at the expense of interpretability and efficiency. Our work aims to bridge this gap by building a lightweight, modular, and adaptive model grounded in basis decomposition theory.

**Statistical methods.** Classical approaches such as ARIMA, SARIMA, and Exponential Smoothing Box et al. (2015); Hyndman & Athanasopoulos (2008) remain popular due to their interpretability and simplicity. These models excel at capturing stationary trends and seasonality, but struggle with nonlinear or nonstationary signals. In addition, their assumptions about data distributions and reliance on fixed smoothing mechanisms make them less effective in the presence of local anomalies or abrupt changes.

---

[0]RAYQUAZA, a legendary Pokémon known for its ability to maintain balance in the Pokémon world, embodies power and adaptability across scales. It reflects the multi-scale nature of our new network.

**Neural models.** Deep learning has expanded the modeling capacity for complex and noisy time-series. RNNs Hochreiter & Schmidhuber (1997) and variants like LSTMs and GRUs capture sequential dependencies, but suffer from gradient vanishing and training inefficiencies. TCNs Bai et al. (2018) leverage dilated convolutions for longer-range memory, but remain limited by their fixed receptive fields. DeepAR Salinas et al. (2020) offers probabilistic sequence modeling, yet suffers from error accumulation over long horizons.

**Transformer-based models.** Inspired by breakthroughs in NLP, transformers have gained traction in forecasting. Informer Zhou et al. (2021) improves efficiency through sparse attention, while FEDformer Zhou et al. (2022b) combines temporal and frequency views using Fourier decomposition. PatchTST Nie et al. (2023a) introduces patching strategies from vision transformers to reduce computational cost. Despite their accuracy, such models are typically large and computationally expensive, making them unsuitable for low-resource or real-time scenarios.

**Lightweight architectures.** Several recent efforts aim to reduce model complexity while retaining competitive performance. The N-BEATS model Oreshkin et al. (2019a) builds modular trend and seasonality blocks using fully connected residual stacks. DLinear Zeng et al. (2022) simplifies forecasting via linear decomposition, outperforming larger models in many cases. More recently, ultra-lightweight models such as OLS-based linear forecasters Toner & Darlow (2024) and FITS Xu et al. (2024b) have demonstrated surprisingly strong performance with minimal parameters. While these models are highly efficient, they often rely on fixed basis templates or linear assumptions, which can limit their ability to capture input-specific or localized structures.

**Basis function methods.** An alternative to learning dense latent representations is to decompose a signal into a combination of basis functions. This has a long history in signal processing with fixed bases such as the Fourier and Wavelet Transforms. In deep learning, N-BEATS Oreshkin et al. (2019a) follows this tradition by learning a set of polynomial and seasonal basis functions. However, this learned basis is globally fixed for all samples, limiting its adaptability. More recent work has introduced greater flexibility. For instance, BasisFormer Ni et al. (2024) learns a set of global bases and then uses an attention mechanism to find an adaptive combination for each forecast. While the combination is adaptive, the underlying basis functions remain global and fixed. In contrast, our work introduces a new level of flexibility, where the basis functions themselves are dynamically generated for each input sample. Using an input-conditioned hypernetwork, **RAYQUAZA** learns a unique and specialized basis of Gaussian atoms, enabling a more parsimonious and expressive representation of non-stationary phenomena.

**Hypernetwork-based methods.** Hypernetworks Ha et al. (2016) introduce an auxiliary network that generates the parameters of a target model, enabling data- or task-dependent specialization. A recent survey Chauhan et al. (2024) provides a taxonomy based on the hypernetwork inputs, the structure of generated parameters, and the degree of per-sample variability. Lightweight variants restrict the hypernetwork to producing low-dimensional or structured parameters, which naturally aligns with basis-function construction in time-series forecasting. Whereas models such as BasisFormer compute input-conditioned weights over a fixed global dictionary, **RAYQUAZA** generates the basis functions themselves: a compact hypernetwork produces per-sample Gaussian atoms, positioning the method within the class of structured, low-dimensional hypernetworks tailored to sequence-specific dynamics.

## 3 RAYQUAZA

**RAYQUAZA** is a novel, modular, and parameter-efficient architecture designed to learn an **adaptive and interpretable representation** of univariate time-series. It processes the input sequence through three complementary modules, each capturing a distinct component of the signal's structure. These component representations are then aggregated through a shallow fusion layer to generate the final forecast. The architecture comprises the following modules:

- **Input-Conditioned Radial Basis Function (iRBF).** iRBF generates Gaussian basis functions whose centers, widths, and amplitudes are predicted per input, enabling dynamic, localized decomposition of transient behaviors such as spikes or shifts.

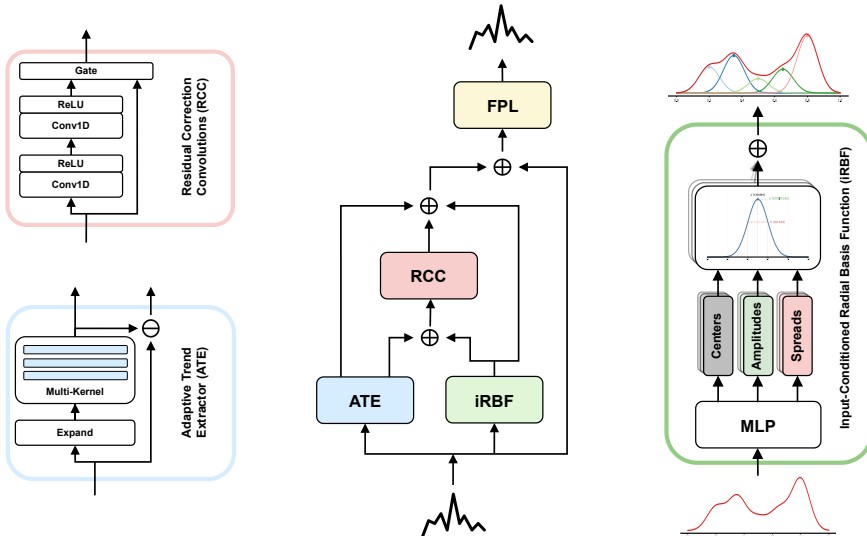

Figure 2: Architecture of **RAYQUAZA**, a lightweight and modular forecasting model. It captures multi-scale structure using four key components: ATE for global patterns, iRBF for localized decomposition, RCC for high-frequency refinement, and FPL for output integration.

- **Adaptive Trend Extractor (ATE).** ATE applies multiple softmax-normalized smoothing filters to extract low-frequency trends and seasonal components.
- **Residual Correction Convolutions (RCC).** RCC captures high-frequency residual signals through stacked gated convolutional blocks with residual connections.
- **Fusion Projection Layer (FPL).** FPL combines the outputs of iRBF, ATE, RCC, and a gated residual connection to the raw input, then projects the sum to the final forecast.

**RAYQUAZA** architectural overview is shown in Figure 2.

## 3.1 INPUT-CONDITIONED RADIAL BASIS FUNCTION (IRBF) MODULE

The iRBF module is designed to learn a representation of sharp transient behaviors such as spikes, dips, or sudden transitions. Unlike classical decompositions with fixed or globally optimized bases, the iRBF layer learns localized basis functions conditioned on each input sequence, enabling it to represent non-stationary dynamics that are often missed by globally parameterized models. While classical RBF networks Park & Sandberg (1991); Poggio & Girosi (1990) use a static set of basis functions, the iRBF module dynamically generates the parameters for a new basis for every sample. This allows for a tailored, per-instance decomposition of each sequence. To our knowledge, this is the first use of a per-input RBF decomposition in a neural forecasting model.

Let B denote the batch size and L the sequence length. Given an input $\mathbf{x} \in \mathbb{R}^{B \times L}$, a multilayer perceptron generates $K$ basis functions by predicting:

$$[\mathbf{c}, \log \boldsymbol{\sigma}, \boldsymbol{\alpha}] = \mathrm{MLP}_{\mathrm{iRBF}}(\mathbf{x}), \tag{1}$$

where $\mathbf{c}, \log \boldsymbol{\sigma}, \boldsymbol{\alpha} \in \mathbb{R}^{B \times K}$ represent the predicted centers, log-scales, and amplitudes, respectively. To enforce the positivity of the scales, the exponential function is applied :

$$\boldsymbol{\sigma} = \exp(\log \boldsymbol{\sigma}) \tag{2}$$

Each basis function defines a Gaussian response at time index $\ell \in \{0, \dots, L-1\}$:

$$\phi_{b,k}(\ell) = \alpha_{b,k} \exp\left(-\frac{1}{2}\left(\frac{\ell - c_{b,k}}{\sigma_{b,k}}\right)^2\right) \tag{3}$$

The output of the iRBF module aggregates all $K$ components:

$$\mathbf{y}_{\text{iRBF}}(b, \ell) = \sum_{k=1}^{K} \phi_{b,k}(\ell) \tag{4}$$

Each basis function corresponds to a localized structure in the signal, offering fine-grained control and a direct path to interpretability. As we demonstrate in Section 5.6 (see Figure 3), the placement and shape of these learned atoms can be directly inspected, providing valuable insight into the model's reasoning.

## 3.2 ADAPTIVE TREND EXTRACTOR (ATE) MODULE

ATE targets low-frequency structures such as trends and seasonality through a learnable ensemble of smoothing filters. Given an input $\mathbf{x} \in \mathbb{R}^{B \times L}$, the module applies $H$ convolutional filters of length $p$, with raw weights $\mathbf{w}_{\text{raw}} \in \mathbb{R}^{H \times 1 \times p}$. To enforce smoothing, weights are normalized using softmax along the kernel dimension:

$$\mathbf{w}[h, 0, :] = \text{softmax}(\mathbf{w}_{\text{raw}}[h, 0, :]), \quad \forall h \in \{1, \ldots, H\} \tag{5}$$

The input is reshaped as $\mathbf{x}_{\text{3D}} \in \mathbb{R}^{B \times 1 \times L}$ and convolved with the normalized filters:

$$\mathbf{y}_{\text{3D}} = \text{Conv1D}(\mathbf{x}_{\text{3D}}, \mathbf{w}), \quad \mathbf{y}_{\text{3D}} \in \mathbb{R}^{B \times H \times L} \tag{6}$$

A trainable combination vector $\boldsymbol{\alpha} \in \mathbb{R}^{H}$, normalized via softmax, aggregates the filtered outputs:

$$\mathbf{y}_{\text{ATE}}(b, \ell) = \sum_{h=1}^{H} \alpha_h \, \mathbf{y}_{\text{3D}}(b, h, \ell) \tag{7}$$

This yields a smoothed representation $\mathbf{y}_{\text{ATE}} \in \mathbb{R}^{B \times L}$ that retains the large-scale structure of the signal.

## 3.3 RESIDUAL CORRECTION CONVOLUTIONS (RCC) MODULE

Although the ATE and iRBF modules capture global and localized behaviors, residual structures may still remain. To address this, we introduce the RCC module to refine these residual components. Each RCC block comprises a shallow 1D convolution followed by a pointwise (1×1) projection and a residual skip connection:

$$\mathbf{y}_1 = \text{ReLU}(\text{Conv1D}(\mathbf{x}, k)), \quad \mathbf{x}_{\text{RCC}} = \mathbf{x} + \text{Conv1D}(\mathbf{y}_1, 1) \tag{8}$$

RCC blocks are computationally lightweight and effective in modeling high-frequency variations and subtle patterns not captured by the ATE or iRBF modules. Multiple RCC modules can be stacked based on the complexity of the dataset and forecast horizon.

## 3.4 FUSION PROJECTION LAYER (FPL)

FPL integrates the outputs of all preceding modules to form the final prediction. Specifically, it sums the outputs of the ATE, iRBF, and RCC modules, along with a gated residual connection to the raw input ($\mathbf{y}_{\text{fusion}}$):

$$\mathbf{y}_{\text{fusion}} = \mathbf{y}_{\text{ATE}} + \mathbf{y}_{\text{iRBF}} + \mathbf{x}_{\text{RCC}} + \gamma \mathbf{x} \tag{9}$$

where $\gamma$ is a learnable scalar that gates the contribution of the original input. The fused signal is passed through a lightweight MLP ($\mathbf{y}_{\text{final}}$):

$$\mathbf{y}_{\text{final}} = \text{MLP}(\mathbf{y}_{\text{fusion}}) \tag{10}$$

which produces the forecasted sequence. The model is trained end-to-end using Mean Squared Error (MSE) loss. The FPL supports non-linear interactions among components while maintaining parameter efficiency.

## 4 EXPERIMENT

### 4.1 DATASETS

**M4 dataset.** It comprises 100,000 univariate time-series sampled from diverse real-world domains, including economic, financial, industrial, and demographic sectors. Each series has a different

frequency (yearly, quarterly, monthly, etc.), making M4 a heterogeneous benchmark for short-term forecasting. Unlike standard long-term forecasting datasets, where samples are generated using sliding windows from a single sequence, the M4 dataset consists of samples originating from independent sources. This introduces significant variability in scale, trend, and periodicity, requiring models to generalize across disparate dynamics. We follow the official competition protocol Oreshkin et al. (2019a) and evaluate on the canonical split. A statistical summary of M4 is provided in Appendix A Table 6.

**ETT (Electricity Transformer Temperature) datasets.** They are derived from sensors deployed on electric transformers across two counties in China. Each variant records seven features (e.g., load, oil temperature) at two different resolutions: 15 minutes (m) and 1 hour (h). This yields four datasets: ETTh1, ETTh2, ETTm1, and ETTm2. In this work, we adopt the long-term univariate forecasting setup and use only the target variable (oil temperature), as established in recent benchmarks Nie et al. (2023a); Zeng et al. (2022). We fix the input sequence length to 336 and report performance on the prediction horizons $T \in \{96, 192, 336, 720\}$. Compared to M4, ETT provides longer, regularly sampled sequences, enabling the study of fine-grained temporal extrapolation. ETT statistics are summarized in Table 6.

**TFB (Time-series Forecasting Benchmark) dataset.** Introduced by Qiu et al. (2024), the TFB is a large-scale and comprehensive benchmark designed to facilitate fair and standardized evaluation of forecasting models. It contains a diverse collection of time-series from numerous real-world domains such as finance, economics, and nature, spanning six distinct frequencies from hourly to yearly. This heterogeneity makes it a robust test for model generalization. We follow the official competition protocol and report performance using the Mean Symmetric Mean Absolute Percentage Error (MS-MAPE) metric.

## 4.2 BASELINES

We compare **RAYQUAZA** against a broad range of baselines to validate its performance across the full spectrum of complexity. These baselines include ultra-lightweight models such as OLS Toner & Darlow (2024) and FITS Xu et al. (2024b), as well as prominent deep architectures spanning diverse paradigms. Key competitors include transformer-based models (PatchTST Nie et al. (2023a), FEDformer Zhou et al. (2022b)), modern convolutional networks (ModernTCN Li et al. (2024), TimesNet Wu et al. (2023)), popular linear-based models (DLinear Zeng et al. (2022)), and modular decomposition frameworks (N-BEATS Oreshkin et al. (2019a)). This selection enables a comprehensive evaluation of **RAYQUAZA**'s accuracy and parameter efficiency.

## 4.3 EXPERIMENTAL SETTINGS

To ensure a fair comparison, all baseline models are trained using the official TimesNet configuration, without modification. **RAYQUAZA** uses the same training protocol, with architecture-specific settings applied uniformly across datasets. Full implementation and training details are provided in the Appendix A.

## 5 RESULTS

We conduct a comprehensive set of experiments to validate **RAYQUAZA**'s performance, following a narrative of escalating evidence. We first establish its strong performance on classic short- and long-term benchmarks (M4 and ETT). We then demonstrate its state-of-the-art capability on the modern, large-scale TFB benchmark. Finally, we confirm its unique expressive power through a rigorous channel-wise evaluation against lightweight linear baselines.

## 5.1 PERFORMANCE ON THE M4 BENCHMARK

We begin by evaluating **RAYQUAZA** on the classic M4 benchmark, which comprises 100,000 diverse short-term time-series. The full results on the M4 benchmark are presented in Table 1. RAYQUAZA demonstrates consistently strong performance, achieving the best OWA score in the Monthly and Others categories and the lowest overall Weighted Average OWA of 0.835, outperforming all baselines.

Table 1: Performance comparison of different models across datasets and metrics. Red values indicate the best results for each dataset and metric.

| Models | Yearly | | | Quarterly | | | Monthly | | | Others | | | Weighted Avg. | | |
|---|---|---|---|---|---|---|---|---|---|---|---|---|---|---|---|
| | SMAPE | MASE | OWA | SMAPE | MASE | OWA | SMAPE | MASE | OWA | SMAPE | MASE | OWA | SMAPE | MASE | OWA |
| N-BEATS (2019) | 13.436 | 3.043 | 0.794 | 10.124 | 1.169 | 0.886 | 12.667 | 0.937 | 0.880 | 4.925 | 3.391 | 1.053 | 11.829 | 1.585 | 0.850 |
| N-HiTS (2022) | 13.418 | 3.045 | 0.793 | 10.202 | 1.194 | 0.899 | 12.791 | 0.969 | 0.899 | 5.061 | 3.216 | 1.040 | 11.927 | 1.613 | 0.861 |
| SCINet (2022) | 13.717 | 3.076 | 0.807 | 10.845 | 1.295 | 0.965 | 13.208 | 0.999 | 0.928 | 5.423 | 3.583 | 1.136 | 12.369 | 1.677 | 0.894 |
| MICN (2023) | 14.935 | 3.523 | 0.900 | 11.452 | 1.389 | 1.026 | 13.773 | 1.076 | 0.983 | 6.716 | 4.717 | 1.451 | 13.130 | 1.896 | 0.980 |
| TimesNet (2021) | 13.387 | 2.996 | 0.786 | 10.100 | 1.182 | 0.894 | 12.667 | 0.937 | 0.880 | 4.891 | 3.302 | 1.035 | 11.851 | 1.599 | 0.855 |
| DLinear (2022) | 16.965 | 4.283 | 1.058 | 12.145 | 1.520 | 1.106 | 13.514 | 1.037 | 0.956 | 6.709 | 4.953 | 1.487 | 13.639 | 2.095 | 1.051 |
| LightTS (2022) | 14.247 | 3.109 | 0.827 | 11.364 | 1.328 | 1.000 | 14.014 | 1.053 | 0.981 | 15.880 | 11.434 | 3.474 | 13.252 | 2.111 | 1.051 |
| Informer (2021) | 14.727 | 3.418 | 0.881 | 11.360 | 1.401 | 1.027 | 14.062 | 1.141 | 1.027 | 24.460 | 20.960 | 5.879 | 14.086 | 2.718 | 1.230 |
| MTS-mixer (2023) | 13.548 | 3.091 | 0.803 | 10.128 | 1.196 | 0.896 | 12.717 | 0.931 | 0.879 | 4.817 | 3.255 | 1.020 | 11.892 | 1.608 | 0.859 |
| RLinear (2023) | 13.994 | 3.015 | 0.807 | 10.702 | 1.299 | 0.959 | 13.363 | 1.014 | 0.940 | 5.437 | 3.706 | 1.157 | 12.473 | 1.677 | 0.898 |
| RMLP (2023) | 13.418 | 3.006 | 0.789 | 10.382 | 1.234 | 0.921 | 12.998 | 0.976 | 0.909 | 5.098 | 3.364 | 1.067 | 12.072 | 1.624 | 0.870 |
| FEDformer (2022) | 13.728 | 3.048 | 0.803 | 10.792 | 1.283 | 0.958 | 14.260 | 1.102 | 1.012 | 4.954 | 3.264 | 1.036 | 12.840 | 1.701 | 0.918 |
| Crossformer (2023) | 13.392 | 3.001 | 0.787 | 16.317 | 2.197 | 1.542 | 12.924 | 0.966 | 0.902 | 5.493 | 3.690 | 1.160 | 13.474 | 1.866 | 0.985 |
| PatchTST (2023) | 13.258 | 2.985 | 0.781 | 10.179 | 1.212 | 0.904 | 12.641 | 0.930 | 0.876 | 4.946 | 2.985 | 1.044 | 11.807 | 1.590 | 0.851 |
| CARD (2023) | 13.302 | 3.016 | 0.786 | 10.179 | 1.176 | 0.884 | 12.670 | 0.933 | 0.878 | 5.330 | 3.261 | 1.075 | 11.815 | 1.587 | 0.850 |
| ModernTCN (2024) | 13.226 | 2.957 | 0.777 | 9.971 | 1.167 | 0.878 | 12.556 | 0.917 | 0.866 | 4.715 | 3.107 | 0.986 | 11.698 | 1.556 | 0.838 |
| SKOLR (2025) | 13.291 | 2.996 | 0.784 | 9.986 | 1.166 | 0.878 | 12.536 | 0.921 | 0.867 | 4.652 | 3.233 | 0.999 | 11.704 | 1.572 | 0.843 |
| Koopa (2023) | 13.352 | 2.997 | 0.786 | 10.159 | 1.189 | 0.901 | 12.730 | 0.953 | 0.901 | 5.124 | 3.124 | 1.004 | 11.863 | 1.595 | 0.858 |
| **RAYQUAZA (ours)** | 13.259 | 2.966 | 0.779 | 9.962 | 1.163 | 0.874 | 12.483 | 0.914 | 0.862 | 4.630 | 3.117 | 0.979 | 11.664 | 1.556 | 0.835 |

*The original N-BEATS paper Oreshkin et al. (2019b) uses an ensemble strategy. For fairness, we only report single-model results.

## 5.2 PERFORMANCE ON LONG-HORIZON BENCHMARKS

Next, we evaluate long-horizon forecasting performance in ETT datasets, following the established univariate protocol used in prior work Li et al. (2024); Oreshkin et al. (2019b); Nie et al. (2023b). As shown in the detailed results in Table 2, RAYQUAZA consistently achieves top-tier performance across all horizons. It secures the most first-place wins (17 out of 32 settings), demonstrating a particular advantage in longer-horizon forecasting on the ETTh1 and ETTh2 datasets while remaining competitive across all scenarios. This performance leads to the best overall average MSE among all compared models.

Table 2: Univariate long-term forecasting on ETT. Input length = 336, horizons $T \in \{96, 192, 336, 720\}$. First-place (min) MSE in each row is red, and the bottom row shows the total $1^{st}$-place counts.

| Models | **RAYQUAZA Ours** | | ModernTCN (2024) | | PatchTST (2023) | | DLinear (2022) | | FEDformer (2022) | | Autoformer (2021) | | Informer (2021) | | LogTrans (2019a) | |
|---|---|---|---|---|---|---|---|---|---|---|---|---|---|---|---|---|
| Metric | MSE | MAE | MSE | MAE | MSE | MAE | MSE | MAE | MSE | MAE | MSE | MAE | MSE | MAE | MSE | MAE |
| ETTh1 96 | 0.055 | 0.179 | 0.055 | 0.179 | 0.055 | 0.179 | 0.056 | 0.180 | 0.079 | 0.215 | 0.071 | 0.206 | 0.193 | 0.377 | 0.283 | 0.468 |
| ETTh1 192 | 0.067 | 0.204 | 0.070 | 0.205 | 0.071 | 0.205 | 0.071 | 0.204 | 0.104 | 0.245 | 0.114 | 0.262 | 0.217 | 0.395 | 0.234 | 0.409 |
| ETTh1 336 | 0.080 | 0.226 | 0.074 | 0.214 | 0.076 | 0.220 | 0.098 | 0.244 | 0.119 | 0.270 | 0.107 | 0.258 | 0.202 | 0.381 | 0.386 | 0.546 |
| ETTh1 720 | 0.082 | 0.230 | 0.086 | 0.232 | 0.087 | 0.236 | 0.087 | 0.359 | 0.142 | 0.299 | 0.126 | 0.283 | 0.183 | 0.355 | 0.475 | 0.629 |
| ETTh2 96 | 0.122 | 0.274 | 0.124 | 0.275 | 0.129 | 0.282 | 0.131 | 0.279 | 0.128 | 0.271 | 0.153 | 0.306 | 0.213 | 0.373 | 0.217 | 0.379 |
| ETTh2 192 | 0.161 | 0.323 | 0.164 | 0.321 | 0.168 | 0.328 | 0.176 | 0.329 | 0.185 | 0.330 | 0.204 | 0.351 | 0.227 | 0.387 | 0.281 | 0.429 |
| ETTh2 336 | 0.179 | 0.344 | 0.171 | 0.336 | 0.171 | 0.336 | 0.209 | 0.367 | 0.231 | 0.378 | 0.246 | 0.389 | 0.242 | 0.401 | 0.293 | 0.437 |
| ETTh2 720 | 0.208 | 0.367 | 0.228 | 0.384 | 0.223 | 0.380 | 0.276 | 0.426 | 0.278 | 0.420 | 0.268 | 0.409 | 0.291 | 0.439 | 0.218 | 0.387 |
| ETTm1 96 | 0.026 | 0.123 | 0.026 | 0.121 | 0.026 | 0.121 | 0.028 | 0.123 | 0.033 | 0.140 | 0.056 | 0.183 | 0.109 | 0.277 | 0.049 | 0.171 |
| ETTm1 192 | 0.039 | 0.150 | 0.040 | 0.152 | 0.039 | 0.150 | 0.045 | 0.156 | 0.058 | 0.186 | 0.081 | 0.216 | 0.151 | 0.310 | 0.157 | 0.317 |
| ETTm1 336 | 0.050 | 0.173 | 0.053 | 0.173 | 0.053 | 0.173 | 0.061 | 0.182 | 0.084 | 0.231 | 0.076 | 0.218 | 0.427 | 0.591 | 0.289 | 0.459 |
| ETTm1 720 | 0.071 | 0.203 | 0.073 | 0.206 | 0.073 | 0.206 | 0.080 | 0.210 | 0.102 | 0.250 | 0.110 | 0.267 | 0.438 | 0.586 | 0.430 | 0.579 |
| ETTm2 96 | 0.067 | 0.192 | 0.065 | 0.183 | 0.065 | 0.186 | 0.063 | 0.183 | 0.067 | 0.198 | 0.065 | 0.189 | 0.088 | 0.225 | 0.075 | 0.208 |
| ETTm2 192 | 0.098 | 0.236 | 0.095 | 0.232 | 0.093 | 0.231 | 0.092 | 0.227 | 0.102 | 0.245 | 0.118 | 0.256 | 0.132 | 0.283 | 0.129 | 0.275 |
| ETTm2 336 | 0.125 | 0.271 | 0.119 | 0.261 | 0.120 | 0.265 | 0.119 | 0.261 | 0.130 | 0.279 | 0.154 | 0.305 | 0.180 | 0.336 | 0.154 | 0.302 |
| ETTm2 720 | 0.175 | 0.327 | 0.173 | 0.323 | 0.171 | 0.323 | 0.175 | 0.320 | 0.178 | 0.325 | 0.182 | 0.335 | 0.300 | 0.435 | 0.160 | 0.321 |
| $1^{st}$ Count | **17** | | 12 | | 10 | | 8 | | 0 | | 0 | | 0 | | 0 | |

## 5.3 EVALUATION ON A LARGE-SCALE BENCHMARK

To validate our performance on a modern, large-scale benchmark, we evaluate **RAYQUAZA** on TFB. As shown in Table 3, **RAYQUAZA** achieves the best MS-MAPE score in all six frequencies against a representative set of prominent models, confirming its state-of-the-art capability. The full benchmark results are in the Appendix B.1.

Table 3: Summary of MS-MAPE on the TFB Benchmark. Lower is better.

| Model | Hourly | Daily | Weekly | Monthly | Quarterly | Yearly |
|---|---|---|---|---|---|---|
| DLinear (2022) | 27.319 | 26.792 | 50.164 | 16.285 | 18.070 | 23.540 |
| FEDformer (2022a) | 28.480 | 25.422 | 53.293 | 14.678 | 16.813 | 23.864 |
| Xgboost (2016) | 28.921 | 25.923 | 36.482 | 14.903 | 16.348 | 21.333 |
| **RAYQUAZA (ours)** | 26.161 | 20.547 | 18.097 | 11.304 | 15.394 | 20.137 |

## 5.4 Channel-Wise Evaluation

Finally, to address the limitations of single-variable protocols and to specifically test our model's expressive power, we conducted a rigorous channel-wise evaluation. Table 4 compares **RAYQUAZA** against ultra-lightweight linear baselines in the ETT and Weather datasets for short (T=96) and long (T=720) horizons. The results show that while linear models are competitive in some settings, **RAYQUAZA demonstrates a decisive advantage on complex long-horizon tasks**. This is particularly evident in the volatile ETTh2 dataset, where its long-horizon error is up to **3x lower** than the baselines, confirming the critical role of its adaptive, non-linear representation. The complete results across all horizons and datasets are in the Appendix B.2.

Table 4: Channel-wise MSE on ETT and Weather datasets. Lower is better. Results for short (T=96) and long (T=720) horizons are shown. Full results are in the appendix.

| Dataset | Horizon | OLS( 2024) | FITS( 2024b) | RAYQUAZA (Ours) |
|---|---|---|---|---|
| ETTh1 | 96 | 0.376 | 0.378 | 0.374 |
| | 720 | 0.491 | 0.506 | 0.480 |
| ETTh2 | 96 | 0.309 | 0.307 | 0.298 |
| | 720 | 0.900 | 0.971 | 0.313 |
| Weather | 96 | 0.142 | 0.144 | 0.138 |
| | 720 | 0.304 | 0.304 | 0.301 |

## 5.5 Comparative Advantage

**RAYQUAZA** achieves consistent top-tier performance in both short- and long-term forecasting tasks, outperforming transformer-based models such as PatchTST and FEDformer, which often exhibit dataset-specific strengths but lack robustness across domains. This generalization is enabled by a modular design that combines localized decomposition, adaptive trend extraction, and residual correction. In particular, on the ETT benchmarks, **RAYQUAZA** matches or exceeds the SOTA performance on longer horizons while maintaining computational efficiency, demonstrating its ability to scale to volatile and high-resolution series without architectural tuning. These results position **RAYQUAZA** as a practical, general-purpose solution for time-series forecasting.

## 5.6 Ablation: Basis–Family Study on M4-Yearly

A core idea of **RAYQUAZA** is that **per-input** localization is more important than expanding the basis dictionary. To test this, we replace the iRBF with four canonical basis families and train each variant under **identical** optimization schedules on 1,000 randomly selected M4-YEARLY series in the Appendix D.

We compare iRBF against four alternative basis families. **FixedRBF** Broomhead & Lowe (1988) uses eight Gaussian bumps with hand-set centers and widths, where only amplitudes are learned. **GlobalRBF** Poggio & Girosi (1990); Park & Sandberg (1991) learns centers, widths, and amplitudes but shares them across all series. **FFT** Box et al. (2015) applies 10 sine–cosine frequency pairs as a standard seasonal basis. **Wavelet** Mallat & Stéphane (1999) uses Daubechies-4 at level 2 to capture multiscale edges and discontinuities.

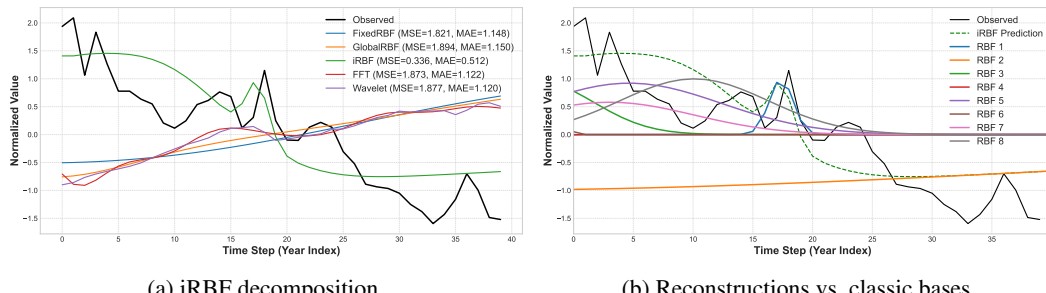

(a) iRBF decomposition.  (b) Reconstructions vs. classic bases.

Figure 3: **Input-conditioned localisation drives the gain.** iRBF places its Gaussians **where** the data require them capturing both the isolated spike and smooth trend, while fixed dictionaries either miss the spike or introduce oscillatory artefacts.

**Qualitative Insight.** Figure 3 visualizes a representative series. Panel (a) shows iRBF assigning a narrow Gaussian exactly at the sharp spike near time index 19, while broader kernels model the slower trend. Panel (b) overlays reconstructions from all baselines: fixed and global Gaussians miss the spike, FFT introduces ringing, and wavelets oversmooth. Only iRBF captures both the high-frequency anomaly and the underlying trend, in line with its lower reconstruction error.

**Quantitative Results.** Table 5 shows that iRBF reduces test MSE from $\approx 1.8$ to **0.336** and MAE from $\approx 1.1$ to **0.512**, an order-of-magnitude improvement over fixed and global Gaussians, and a 55–60% drop relative to FFT and wavelets, despite all variants using a comparable number of parameters. These results suggest that **where** basis functions are placed, per series, matters more than the complexity or richness of the basis family. A comprehensive qualitative and quantitative analysis of the iRBF representation learned is provided in the Appendix E.

Table 5: Masked reconstruction error on 1,000 M4-YEARLY series. Lower is better. **Bold** marks the best.

| Basis Module | MSE | MAE |
|---|---|---|
| FixedRBF (hand-set) | 1.821 | 1.148 |
| GlobalRBF (shared) | 1.894 | 1.150 |
| FFT (10 freq. pairs) | 1.873 | 1.122 |
| Wavelet (db4, level 2) | 1.877 | 1.120 |
| **iRBF (ours)** | 0.336 | 0.512 |

### 5.7 MODEL EFFICIENCY

**Parameter Efficiency. v** exhibits exceptional parameter efficiency while delivering competitive and often superior forecasting performance, as shown in Figure 1 and Table 11. On the M4-YEARLY dataset, it uses just 55,131 parameters, which is less than 0.5% of ModernTCN (33.9M) and Informer (11.3M), and approximately 9.4% of TimesNet (586K). On the M4-QUARTERLY dataset, it requires only 60,533 parameters, less than 0.1% of ModernTCN (67.7M), 0.5% of Informer, 4.4% of CARD (1.37M), and 1.3% of TimesNet (4.7M). For the M4-MONTHLY dataset, it uses 108,075 parameters under 1% of Informer, 7.7% of CARD (1.41M), and 9.2% of TimesNet (1.17M). On the M4-OTHERS dataset, it maintains an average of 116,653 parameters, which is less than 0.7% of ModernTCN (16.2M), 1% of Informer, 7.6% of CARD (1.54M), and 13.2% of TimesNet (882K). A detailed analysis is provided in the Appendix G.

**Training Time Efficiency. RAYQUAZA** also demonstrates significant gains in training speed. It requires only 6.12% of the per-epoch training time of ModernTCN, reducing the runtime by 3,319.28 seconds on average. Compared to TimesNet, it uses just 22.03% of the per-epoch time, saving 71.29 seconds. These savings persist across the full training schedule, making **RAYQUAZA** well-suited for resource-constrained or time-sensitive applications (see Table 11).

**Impact of iRBF.** The inclusion of the iRBF module is critical to **RAYQUAZA**'s performance, leading to the most significant gains among all components. As shown in the full architectural ablation study in the Appendix C, removing the iRBF module causes the largest degradation in overall performance, increasing the average OWA from **0.835 to 0.866 (+3.46% degradation)**. This drop is consistent across all data frequencies and underscores the importance of the iRBF's ability to adaptively model localized, irregular temporal patterns such as spikes, dips, and structural shifts that are difficult to capture with the trend and residual modules alone.

## 6 CONCLUSION

We presented **RAYQUAZA**, a compact architecture that learns to decompose time-series into an adaptive basis of input-conditioned Gaussian atoms. Our comprehensive evaluation showed that this approach achieves state-of-the-art results on large-scale benchmarks and excels where linear models fail, all while remaining exceptionally parameter-efficient. **RAYQUAZA** proves that a focus on expressive, adaptive representations, not just model scale, is a key to unlocking the next generation of efficient and powerful forecasting models.

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

## A    EXPERIMENTAL SETTINGS

**RAYQUAZA Configuration.** RAYQUAZA is designed to adapt to varying temporal structures through four modular components: the input-conditioned RBF (iRBF) module, the Adaptive Trend Extractor (ATE), Residual Correction Convolutions (RCC), and the Fusion Projection Layer (FPL). The number of basis functions in the iRBF module varies with dataset frequency: 8 for Yearly, Quarterly, and Daily; 12 for Weekly and Monthly; and 16 for Hourly. In the ATE module, the number of smoothing kernels $H$ is set to 3 by default, but increased to 4 and 5 for Yearly and Quarterly datasets, respectively. Kernel dilations follow $p = \max(\frac{L}{10}, 5)$, where $L$ is the input sequence length. The RCC module is configured with 2–6 stacked convolutional blocks, kernel sizes of 3, 12, or 64, and channel sizes of 128 or 256, depending on the dataset. The FPL integrates all component outputs and projects them to the final forecast.

**RAYQUAZA Training.** All **RAYQUAZA** experiments are carried out using Adam optimizer with a learning rate of $10^{-3}$, batch size 16, and early stopping based on SMAPE validation. Each model is trained for up to 10 epochs. The implementation is in PyTorch and is executed on a single NVIDIA Quadro RTX GPU with 16 GB of memory.

**Baseline Configuration.** All baseline models (e.g., PatchTST, FEDformer, Informer, Autoformer, DLinear, ModernTCN, CARD) are trained using the official codebase and the default settings provided in the TimesNet repository.[1] Input sequence lengths are fixed to 336 for the ETT datasets, with prediction horizons $\{96, 192, 336, 720\}$. M4 input lengths follow the competition protocol (typically a multiple of the forecast horizon). Data are normalized using training-set z-scores. The models use the Adam optimizer with learning rates of $10^{-5}$ for transformer models and $10^{-3}$ for linear baselines. The batch size is 16 or 32. Early stopping is applied with a patience of 3, and a fixed seed (2021) is used for all runs. Model-specific parameters, such as PatchTST's patch size and stride or FEDformer's frequency-domain attention modules, are kept unchanged. All metrics (SMAPE, MASE, OWA for M4 and MSE, MAE for ETT) are computed on the test set, with no post hoc tuning.

**Evaluation Metrics** We evaluate forecasting quality with several widely used metrics. For the M4 and ETT benchmarks, we use *SMAPE* (Symmetric Mean Absolute Percentage Error) Makridakis (1993), *MASE* (Mean Absolute Scaled Error) Hyndman & Koehler (2006), and *OWA* (Overall Weighted Average) Makridakis et al. (2020), together with the classical *MAE* and *MSE*. For the TFB benchmark, we report its primary metric, *MS-MAPE* (Mean SMAPE) Qiu et al. (2024).

For a single series with a forecast horizon of length $n$ and a seasonal period $m$, and a benchmark set with $N$ series, the metrics are defined as:

$$\text{SMAPE} = \frac{1}{n} \sum_{t=1}^{n} \frac{|y_t - \hat{y}_t|}{(|y_t| + |\hat{y}_t|)/2} \times 100, \tag{11}$$

$$\text{MS-MAPE} = \frac{1}{N} \sum_{i=1}^{N} \text{SMAPE}_i, \tag{12}$$

$$\tag{11}$$

---

[1] https://github.com/thuml/Time-Series-Library

$$\text{MAE} = \frac{1}{n} \sum_{t=1}^{n} |y_t - \hat{y}_t|, \tag{13}$$

$$\text{MASE} = \frac{\frac{1}{n} \sum_{t=1}^{n} |y_t - \hat{y}_t|}{\frac{1}{n-m} \sum_{t=m+1}^{n} |y_t - y_{t-m}|}, \tag{14}$$

$$\text{MSE} = \frac{1}{n} \sum_{t=1}^{n} \left(y_t - \hat{y}_t\right)^2, \tag{15}$$

$$\text{OWA} = \frac{\text{SMAPE}}{\text{SMAPE}_{\text{naive}}} + \frac{\text{MASE}}{\text{MASE}_{\text{naive}}}. \tag{16}$$

Equations equation 11, equation 13–equation 16 are reported as defined in the M4 competition Makridakis et al. (2020). The MS-MAPE metric, defined in Equation equation 12, is the primary evaluation metric for the TFB benchmark Qiu et al. (2024). For all metrics, lower scores indicate better forecasts.

**Datasets** To ensure a comprehensive and robust evaluation of RAYQUAZA, we benchmark our model on three distinct classes of widely used time-series forecasting datasets: the short-term M4 competition dataset, the long-horizon ETT (Electricity Transformer Temperature) datasets, and the large-scale TFB (Time-series Forecasting Benchmark). These datasets span a diverse range of domains, frequencies, and temporal characteristics, providing a rigorous test of model generalization and performance. Table 6 provides a detailed statistical summary of each dataset used in our experiments, including its domain, dimensionality, forecast horizon(s), and the size of the training, validation, and test splits.

Table 6: Summary of benchmark datasets used for short-term (M4) and long-term (ETT) forecasting tasks. The M4 dataset includes diverse univariate series from various domains, while the ETT datasets are derived from sensor readings with different temporal resolutions.

| Task | Dataset | Dim | Horizon(s) | Size (Train/Val/Test) | Domain |
|---|---|---|---|---|---|
| Long-term | ETTm1 | 1 | 96 - 192 - 336 - 720 | (34,465 / 11,521 / 11,521) | Electricity |
| | ETTm2 | 1 | 96 - 192 - 336 - 720 | (34,465 / 11,521 / 11,521) | Electricity |
| | ETTh1 | 1 | 96 - 192 - 336 - 720 | (8,545 / 2,881 / 2,881) | Electricity |
| | ETTh2 | 1 | 96 - 192 - 336 - 720 | (8,545 / 2,881 / 2,881) | Electricity |
| Short-term | M4-Yearly | 1 | 6 | (23,000 / 0 / 23,000) | Demographic |
| | M4-Quarterly | 1 | 8 | (24,000 / 0 / 24,000) | Finance |
| | M4-Monthly | 1 | 18 | (48,000 / 0 / 48,000) | Industry |
| | M4-Weekly | 1 | 13 | (359 / 0 / 359) | Macro |
| | M4-Daily | 1 | 14 | (4,227 / 0 / 4,227) | Micro |
| | M4-Hourly | 1 | 48 | (414 / 0 / 414) | Other |

# B  FULL EXPERIMENTAL RESULTS

This section provides the complete, unabridged results that are summarized in the main paper's Section 5.

## B.1  FULL TFB BENCHMARK RESULTS

Table 7 presents the complete results on the TFB benchmark, showing the performance of **RAYQUAZA** against all models included in the original benchmark. This data supports the summary provided in Table 3.

## B.2  FULL CHANNEL-WISE ETT AND WEATHER RESULTS

Table 4 provides the unabridged, per-horizon results for the channel-wise evaluation on all ETT and Weather datasets. This is the complete data summarized in Table 8.

Table 7: MS-MAPE (Mean Symmetric Mean Absolute Percentage Error) Comparison, ordered chronologically. First-place (min) value in each column is red.

| Model (Year) | Hourly | Daily | Weekly | Monthly | Quarterly | Yearly |
|---|---|---|---|---|---|---|
| LR (Classical) | 28.473 | 26.463 | 56.279 | 17.214 | 16.000 | 41.919 |
| AutoCES (1950s-based) | — | 24.457 | 30.525 | 18.750 | 17.440 | 20.272 |
| KF (1960) | 70.382 | 22.004 | 19.288 | 18.355 | 19.524 | 157.051 |
| ND (Classical) | 50.216 | 24.513 | 37.112 | 22.252 | 18.583 | 20.268 |
| NM (Classical) | 43.165 | 42.843 | 113.559 | 28.820 | 35.988 | 42.554 |
| RNN (1980s) | 70.492 | 112.613 | 106.534 | 192.250 | 198.359 | 197.946 |
| Xgboost (2016) | 28.921 | 25.923 | 36.482 | 14.903 | 16.348 | 21.333 |
| TCN (2018) | 41.590 | 77.606 | 71.959 | 137.981 | 192.939 | 199.599 |
| FEDformer (2022) | 28.480 | 25.422 | 53.293 | 14.678 | 16.813 | 23.864 |
| Stationary (2022-era) | 30.702 | 22.205 | 18.270 | 16.378 | 16.753 | 20.281 |
| Crossformer (2023) | 73.124 | 120.256 | 144.661 | 193.177 | 198.238 | 198.211 |
| DLinear (2023) | 27.319 | 26.792 | 50.164 | 16.285 | 18.070 | 23.540 |
| NLinear (2023) | 29.860 | 24.895 | 40.255 | 21.481 | 23.067 | 22.018 |
| TiDE (2023) | 31.181 | 26.197 | 39.461 | 17.172 | 18.295 | 30.446 |
| Triformer (2023) | 36.421 | 71.532 | 75.392 | 118.475 | 191.489 | 199.307 |
| **Rayquaza (Ours)** | 26.161 | 20.547 | 18.097 | 11.304 | 15.394 | 20.137 |

Table 8: Channel-wise long-term forecasting on ETT and Weather. Each variable is forecasted independently, and the results are averaged. Horizons $T \in \{96, 192, 336, 720\}$. First-place (min) MSE in each row is red, and the bottom row reports the total number of $1^{st}$-place counts.

| Models | | **RAYQUAZA (Ours)** | OLS (2024b) | FITS (2024) | TimesNet (2023) | FEDformer (2022) | Autoformer (2021) | Pyraformer (2022) | Informer (2021) |
|---|---|---|---|---|---|---|---|---|---|
| Metric | | MSE | MSE | MSE | MSE | MSE | MSE | MSE | MSE |
| ETTm1 | 96 | 0.315 | 0.306 | 0.310 | 0.338 | 0.379 | 0.505 | 0.543 | 0.672 |
| | 192 | 0.336 | 0.335 | 0.338 | 0.374 | 0.426 | 0.553 | 0.557 | 0.795 |
| | 336 | 0.359 | 0.364 | 0.367 | 0.410 | 0.445 | 0.621 | 0.754 | 1.212 |
| | 720 | 0.410 | 0.413 | 0.435 | 0.478 | 0.543 | 0.671 | 0.908 | 1.166 |
| ETTm2 | 96 | 0.171 | 0.166 | 0.165 | 0.187 | 0.203 | 0.255 | 0.435 | 0.365 |
| | 192 | 0.226 | 0.228 | 0.225 | 0.249 | 0.269 | 0.281 | 0.730 | 0.533 |
| | 336 | 0.283 | 0.295 | 0.291 | 0.321 | 0.325 | 0.339 | 1.201 | 1.363 |
| | 720 | 0.379 | 0.415 | 0.409 | 0.408 | 0.421 | 0.433 | 3.625 | 3.379 |
| ETTh1 | 96 | 0.374 | 0.376 | 0.378 | 0.384 | 0.375 | 0.395 | 0.407 | 0.827 |
| | 192 | 0.419 | 0.413 | 0.413 | 0.436 | 0.427 | 0.443 | 0.452 | 0.891 |
| | 336 | 0.444 | 0.448 | 0.500 | 0.491 | 0.459 | 0.463 | 0.490 | 0.963 |
| | 720 | 0.480 | 0.491 | 0.506 | 0.521 | 0.484 | 0.499 | 0.521 | 1.037 |
| ETTh2 | 96 | 0.298 | 0.309 | 0.307 | 0.340 | 0.340 | 0.297 | 0.320 | 2.133 |
| | 192 | 0.364 | 0.423 | 0.447 | 0.402 | 0.433 | 0.354 | 0.380 | 2.760 |
| | 336 | 0.445 | 0.540 | 0.566 | 0.452 | 0.508 | 0.413 | 0.430 | 3.233 |
| | 720 | 0.313 | 0.900 | 0.971 | 0.462 | 0.480 | 0.457 | 0.479 | 3.659 |
| Weather | 96 | 0.138 | 0.142 | 0.144 | 0.172 | 0.217 | 0.266 | 0.896 | 0.300 |
| | 192 | 0.183 | 0.185 | 0.188 | 0.219 | 0.276 | 0.307 | 0.622 | 0.598 |
| | 336 | 0.239 | 0.235 | 0.238 | 0.280 | 0.339 | 0.359 | 0.739 | 0.578 |
| | 720 | 0.301 | 0.304 | 0.304 | 0.365 | 0.403 | 0.419 | 1.004 | 1.059 |
| $1^{st}$ Count | | **11** | 4 | 3 | 0 | 0 | 3 | 0 | 0 |

## B.3 RESULT STABILITY AND VARIANCE ANALYSIS

To ensure the robustness of our findings, all M4 experiments were conducted over three different random seeds. Table 9 presents the mean and standard deviation of the OWA scores for **RAYQUAZA** compared to a leading baseline, ModernTCN.

The results demonstrate that **RAYQUAZA**'s performance is highly stable, exhibiting low variance across runs. The consistent and favorable outcomes, particularly on the Quarterly, Monthly, and Others subsets, underscore the reliability of our model's performance gains.

Table 9: Mean and standard deviation of OWA scores on M4 subsets over 3 random seeds. Lower is better.

| Model | Yearly OWA | Quarterly OWA | Monthly OWA | Others OWA | Weighted OWA |
|---|---|---|---|---|---|
| ModernTCN | $0.777 \pm 0.002$ | $0.878 \pm 0.001$ | $0.866 \pm 0.002$ | $0.986 \pm 0.005$ | $0.838 \pm 0.001$ |
| **RAYQUAZA** | $0.779 \pm 0.001$ | **$0.876 \pm 0.001$** | **$0.863 \pm 0.001$** | **$0.979 \pm 0.001$** | **$0.835 \pm 0.001$** |

## C    MODEL ABLATION

To assess the contribution of each architectural component in **RAYQUAZA**, we conduct ablation experiments on the M4 dataset by removing one module at a time, either ATE, RCC, or iRBF, while maintaining all hyperparameters and other components fixed. The results are summarized in Table 10.

**Effect of Removing ATE.** Excluding the ATE module, which is responsible for modeling low-frequency trends and seasonal structure, causes the largest overall performance drop. The global average OWA increases from $0.835$ to $0.860$ (**3%** degradation). The impact is most severe on the high frequency of the OTHERS subset, where the OWA increases from $0.979$ to $1.068$, and the SMAPE increases by +0.411. These results confirm that the multikernel smoothing layer plays a critical role in capturing slow-varying components that cannot be effectively recovered by localized or residual modules alone.

**Effect of Removing RCC.** RCC is designed to refine high-frequency residuals through gated convolutions. Its removal leads to a more modest global increase in OWA of **0.60%**. The main impact appears again in the OTHERS subset, where the OWA increases to $1.001$ and the SMAPE increases slightly to $4.738$. These results suggest that RCC plays an important role in modeling fine-grained signal variations, particularly in volatile or noisy sequences.

**Effect of Removing iRBF.** The iRBF module is responsible for localized input-conditioned decomposition. Ablating it leads to the highest overall degradation: a global increase in OWA of **3.46%**, with consistent performance drops across all splits. Quarterly and Monthly subsets show noticeable increases in OWA of +0.03, and OTHERS again suffers with OWA increasing to $1.013$. This underlines the importance of dynamically placing basis functions to model mid-scale transients, such as shifts, spikes, or structural changes.

**Conclusion.** These ablations validate the frequency-specialized design of **RAYQUAZA**. ATE models long-term, low-frequency structures. iRBF captures localized transients, and RCC addresses residual high-frequency noise. Removing any one of them leads to a measurable performance drop, confirming that each module contributes uniquely to the model's robustness across the diverse spectral regimes of M4.

Table 10: Component ablation on the M4 dataset. We report SMAPE, MASE, and OWA for each frequency split (Yearly, Quarterly, Monthly, Others) and their average. The final column shows relative degradation in average OWA (%) compared to full **RAYQUAZA**. Removing ATE, RCC, or iRBF harms performance to varying degrees, confirming the complementary role of each component.

| | Yearly | | | Quarterly | | | Monthly | | | Others | | | Average | | | Promotion |
|---|---|---|---|---|---|---|---|---|---|---|---|---|---|---|---|---|
| | SMAPE | MASE | OWA | SMAPE | MASE | OWA | SMAPE | MASE | OWA | SMAPE | MASE | OWA | SMAPE | MASE | OWA | (%) |
| **RAYQUAZA** | **13.259** | **2.966** | **0.779** | **9.962** | **1.163** | **0.874** | **12.483** | **0.914** | **0.862** | **4.630** | **3.117** | **0.979** | **11.664** | **1.556** | **0.835** | — |
| w/o ATE | 13.487 | 3.056 | 0.797 | 10.098 | 1.175 | 0.887 | 12.777 | 0.943 | 0.886 | 5.041 | 3.409 | 1.068 | 11.911 | 1.608 | 0.860 | 2.75% |
| w/o RCC | 13.403 | 3.009 | 0.789 | 9.979 | 1.160 | 0.876 | 12.469 | 0.915 | 0.862 | 4.738 | 3.185 | 1.001 | 11.700 | 1.569 | 0.842 | 0.60% |
| w/o RBF | 13.722 | 3.071 | 0.807 | 10.311 | 1.204 | 0.906 | 12.920 | 0.947 | 0.893 | 4.792 | 3.229 | 1.013 | 12.072 | 1.611 | 0.866 | 3.46% |

## D    ABLATION: BASIS FAMILY COMPARISON ON M4-YEARLY

To evaluate the effectiveness of the iRBF module in **RAYQUAZA**, we compare it against four canonical basis families under identical experimental settings. The goal is to test whether **learned, per-series basis placement** is more beneficial than using a richer or fixed dictionary shared across samples.

### D.1 EXPERIMENTAL SETUP

We perform a masked reconstruction task on 1,000 randomly sampled series from the M4-YEARLY dataset. Each model is trained to reconstruct input sequences given partial masking and evaluated using masked MSE and MAE. The dataset is split into 70% training, 15% validation, and 15% test sets. All models use identical training schedules: 400 epochs with early stopping (patience = 25), Adam optimizer with a learning rate of $10^{-3}$ and a weight decay of $10^{-4}$ (or 0 for fixed basis variants). Input sequences are z-scored per series, and the padding is used to match sequence lengths

The following basis models are tested:

1. **FixedRBF.** The centers $c_k^\star$ are placed uniformly on the grid, the widths are fixed to $\sigma_k^\star = T/4$, and only the amplitudes $\alpha_k$ are learned:

$$\hat{x}_{\text{FixedRBF}}(t) \ = \ \sum_{k=1}^{K} \alpha_k \ \exp\Big[-\tfrac{1}{2}\big(\tfrac{t-c_k^\star}{\sigma_k^\star}\big)^2\Big]. \tag{12}$$

2. **GlobalRBF.** All three parameters are learned *once* and shared by the entire dataset:

$$\hat{x}_{\text{GlobalRBF}}(t) \ = \ \sum_{k=1}^{K} \alpha_k \ \exp\Big[-\tfrac{1}{2}\big(\tfrac{t-c_k}{\sigma_k}\big)^2\Big]. \tag{13}$$

3. **FFT (10 frequency pairs).** Ten sine–cosine pairs form the seasonal dictionary; we learn real weights $\beta_f^{\text{sin}}, \beta_f^{\text{cos}}$ and a bias $b$:

$$\hat{x}_{\text{FFT}}(t) \ = \ b \ + \ \sum_{f=1}^{10}\Big[\beta_f^{\text{sin}} \ \sin\big(\tfrac{2\pi ft}{T}\big) \ + \ \beta_f^{\text{cos}} \ \cos\big(\tfrac{2\pi ft}{T}\big)\Big]. \tag{14}$$

4. **Wavelet (Daubechies-4, level 2).** Let $\psi_{j,k}$ denote the db4 mother wavelet scaled by $2^j$ and translated to index $k$. We keep all atoms up to level 2 and fit coefficients $w_{j,k}$:

$$\hat{x}_{\text{Wavelet}}(t) \ = \ \sum_{(j,k)\in\mathcal{I}_2} w_{j,k} \ \psi_{j,k}(t), \tag{15}$$

where $\mathcal{I}_2$ indexes the approximation and detail sub-bands at levels $j \leq 2$.

As shown in Table 5, **iRBF** outperforms all alternative basis families by a wide margin. Compared to the best-performing non-iRBF variant (FFT), iRBF achieves a 55–60% reduction in both MSE and MAE. Compared to fixed or global Gaussians, the improvement is over an order of magnitude in MSE. Notably, all models use a comparable number of parameters, confirming that the gain stems from the **input-conditioned placement** of the basis functions, rather than the parameter count or dictionary complexity.

### D.2 QUALITATIVE INSIGHT

Figure 3 illustrates the basis reconstructions for one representative time series. The iRBF model allocates a narrow Gaussian directly on the sharp spike (near index 19), while broader kernels model the underlying trend. In contrast:

- **FixedRBF** and **GlobalRBF** fail to capture the spike due to rigid or shared center placement.
- **FFT** introduces oscillatory ringing near the anomaly.
- **Wavelet** smooths over the peak entirely.

These observations validate that the iRBF mechanism excels at handling transient and low-frequency structures by adapting to the characteristics of individual series.

### D.3 SUMMARY

This ablation confirms that **adaptive, per-sample basis localization** is critical for capturing complex patterns in real-world time series. Despite comparable model sizes, iRBF dramatically outperforms static or global bases. These findings justify its central role in **RAYQUAZA**'s design and explain its advantage in diverse and non-stationary forecasting benchmarks.

Table 11: Comparison of parameter count and per-epoch training time (in seconds) of RAYQUAZA vs. ModernTCN and TimesNet. Fraction columns indicate the percentage of each competitor's cost.

| Dataset | Model | Params | Frac (%) | Time (s) | Frac (%) |
|---|---|---|---|---|---|
| Yearly | RAYQUAZA | 55,131 | — | 13.42 | — |
| | ModernTCN | 33,878,022 | 0.16 | 56.68 | 23.69 |
| | TimesNet | 586,699 | 9.40 | 504.30 | 2.66 |
| Quarterly | RAYQUAZA | 60,533 | — | 14.07 | — |
| | ModernTCN | 67,708,936 | 0.09 | 133.42 | 10.55 |
| | TimesNet | 4,688,409 | 1.29 | 12191.69 | 0.12 |
| Monthly | RAYQUAZA | 108,075 | — | 40.09 | — |
| | ModernTCN | 17,410,066 | 0.62 | 151.43 | 26.47 |
| | TimesNet | 1,174,543 | 9.20 | 625.22 | 6.41 |
| Others Avg | RAYQUAZA | 116,653 | — | 4.24 | — |
| | ModernTCN | 16,238,560 | 0.72 | 15.46 | 27.43 |
| | TimesNet | 761,990 | 15.31 | 27.71 | 15.32 |

# E  ANALYSIS OF THE iRBF ADAPTIVE BASIS REPRESENTATION

A core contribution of our work is the iRBF module, which learns a per-sample adaptive basis representation. This section provides a detailed qualitative and quantitative analysis of the learned basis functions to validate their adaptability, interpretability, and distinguishability.

## E.1  QUALITATIVE ANALYSIS: INPUT-DEPENDENT SPECIALIZATION

To demonstrate how the iRBF module adapts its basis functions to different signal structures, we conducted a reconstruction experiment on four canonical signal types: a low-frequency sine wave, a high-frequency sine wave, a localized pulse, and a bimodal distribution. Figure 4 visualizes the results.

The key insights are as follows.

- **Low Frequency (Top Row).** The model learns a set of wide, overlapping Gaussian atoms that collectively form a smooth basis, perfectly reconstructing the low-frequency sine wave. The RBF amplitudes are distributed to capture the positive and negative phases of the signal.

- **High Frequency (Second Row).** In response to a high-frequency input, the iRBF learns a basis of narrower, more numerous Gaussian atoms that are tiled across the time axis to capture the rapid oscillations.

- **Localized Pulse (Third Row).** For a sharp, transient event, the iRBF demonstrates its locality by learning a single, dominant, and very narrow Gaussian atom placed exactly at the pulse's location. The other atoms have minimal amplitude, resulting in a highly parsimonious representation.

- **Bimodal Shape (Bottom Row).** The model learns to place distinct sets of basis functions centered around each of the two modes of the input signal, demonstrating its ability to represent multi-modal structures.

These visualizations provide strong qualitative evidence that the iRBF module learns a truly adaptive basis, specializing its representation based on the input signal's content rather than relying on a fixed, one-size-fits-all approach.

## E.2  QUANTITATIVE ANALYSIS: KERNEL DISTINGUISHABILITY AND DIVERSITY

To quantitatively validate that the learned iRBF kernels are distinct and adaptive, we analyzed the statistical properties of the learned basis parameters (centers, scales, and amplitudes) across a test set of time-series.

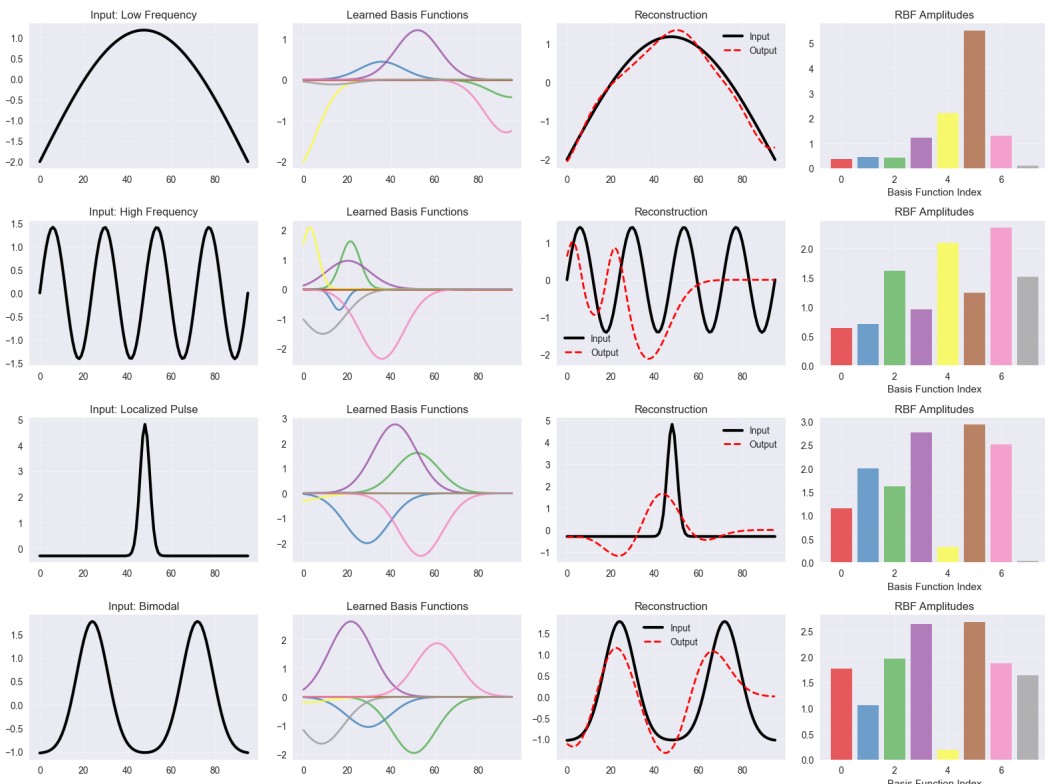

Figure 4: Qualitative analysis of the iRBF module on four canonical signal types. For each signal (row), we show the input, the learned basis functions, the reconstruction, and the amplitudes of the basis functions. The iRBF learns a specialized basis tailored to each input's structure.

**Diversity Metrics.** We define two metrics to measure kernel diversity:

- **Within-Sample Diversity:** The average pairwise distance between the $K$ basis functions for a single input. This measures how different the learned atoms are from each other for a given signal.
- **Between-Sample Diversity:** The average distance of a single basis function (e.g., the first atom) across different input samples. This measures how much a specific atom's role changes from one signal to another.

As visualized in Figure 5, the within-sample diversity is consistently high, confirming that the kernels learned for a given input are well-differentiated. The between-sample diversity is lower but still significant, indicating that kernels adapt their roles across different inputs. Table 12 provides the precise quantitative summary of these findings. The results show negligible to moderate correlations between parameters, suggesting the model is not learning a degenerate or redundant representation. This data confirms that distinguishability in the iRBF module emerges naturally from the MLP's learned ability to dynamically allocate and configure its basis functions based on input content.

# F ADDITIONAL ARCHITECTURAL VALIDATION

To provide a deeper understanding of our design choices and demonstrate the robustness of our results, we present additional ablation studies and experimental details in this section.

## F.1 ABLATION STUDY ON ATE KERNEL NORMALIZATION

A key design choice in our Adaptive Trend Extractor (ATE) module is the use of a softmax function to normalize the convolutional filter weights. We empirically validate this choice by comparing

Table 12: Quantitative summary of iRBF kernel diversity and parameter correlations, averaged over the test set.

| Diversity Metric | Mean $\pm$ SD |
|---|---|
| Within-Sample Center Spread | $15.99 \pm 5.24$ |
| Between-Sample Center Spread | $6.44 \pm 4.07$ |

| Parameter Correlation | Average Value |
|---|---|
| Correlation (Center, Scale) | 0.16 |
| Correlation (Center, Amplitude) | 0.01 |
| Correlation (Scale, Amplitude) | 0.07 |

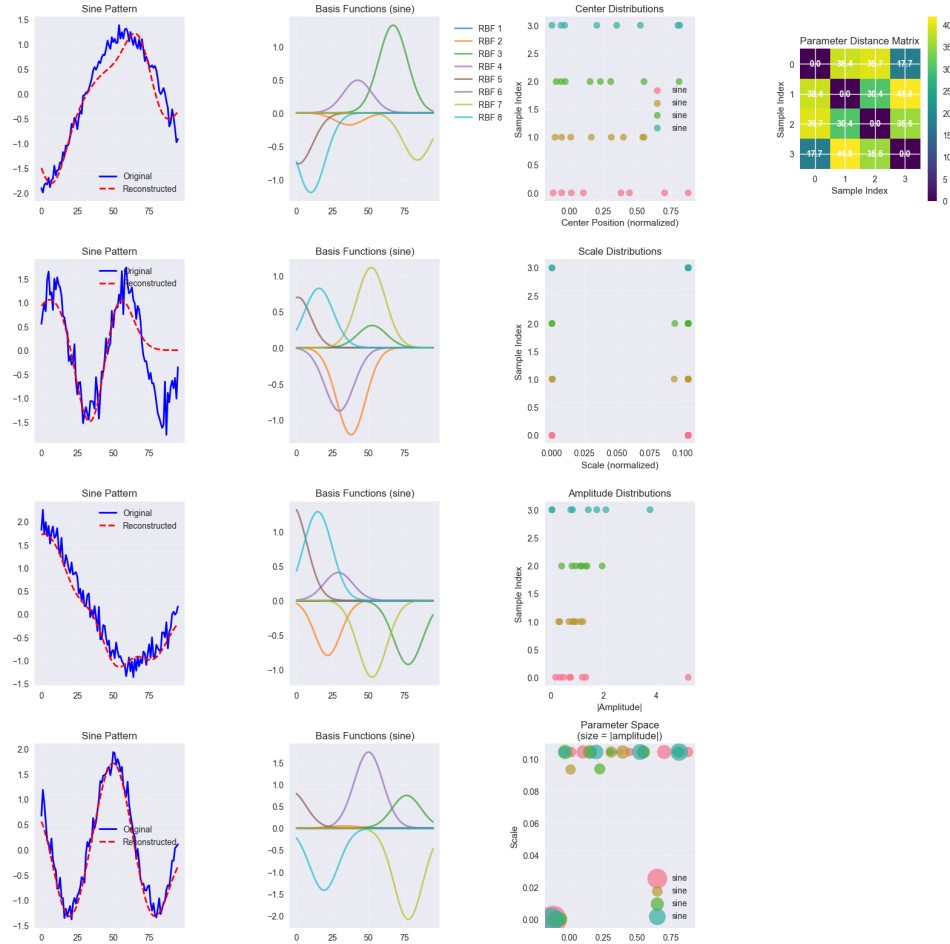

Figure 5: Quantitative visualization of iRBF kernel diversity. The histogram (left) and box plot (right) show that within-sample diversity is significantly higher than between-sample diversity, confirming that kernels are distinct for each input while adapting their roles across different inputs.

three normalization schemes on the M4 benchmark: our proposed softmax, a simple L1-norm, and unconstrained (raw) weights.

As shown in Table 13, the softmax-normalized kernel consistently yields the best performance, achieving the lowest average OWA and converging approximately 30% faster than the alternatives. The softmax function guarantees nonnegative, unit-sum weights, ensuring that the kernel acts as a true low-pass smoother. This prevents undesirable artifacts, such as Gibbs ringing, which we observed with unconstrained kernels.

Although positive-only weights cannot perform differencing, this inductive bias is compensated for by the RCC module, which is specifically designed to capture high-frequency details.

### F.2 HYPERPARAMETER DETAILS

The hyperparameters for **RAYQUAZA** were selected based on a limited search to balance performance and efficiency, rather than an exhaustive tuning process. The search space for the key architectural parameters was as follows:

- **Number of iRBF atoms ($K$):** {4, 8, 12, 16}
- **Number of ATE kernels ($H$):** {2, 3, 4, 5, 6}
- **RCC depth:** {2, 3, 4, 5, 6} with channel sizes in {64, 128, 256}

Standard training hyperparameters were used, including a learning rate of $1 \times 10^{-3}$ and batch sizes of 16 or 32. The total computational cost for this hyperparameter sweep was less than 50% of that reported for tuning comparable models like TimesNet.

Table 13: Comparison of kernel normalization schemes in the ATE module on the M4 benchmark (H=3 kernels, average of 3 seeds).

| Variant | Avg. OWA ↓ | Δ OWA vs. Softmax | Epochs to Converge |
|---|---|---|---|
| Unconstrained | 0.841 | +0.006 | $9.4 \pm 0.4$ |
| L1-norm | 0.839 | +0.004 | $10.0 \pm 0.5$ |
| **Softmax (Ours)** | **0.835** | — | **$7.1 \pm 0.3$** |

## G EXTENDED MODEL EFFICIENCY

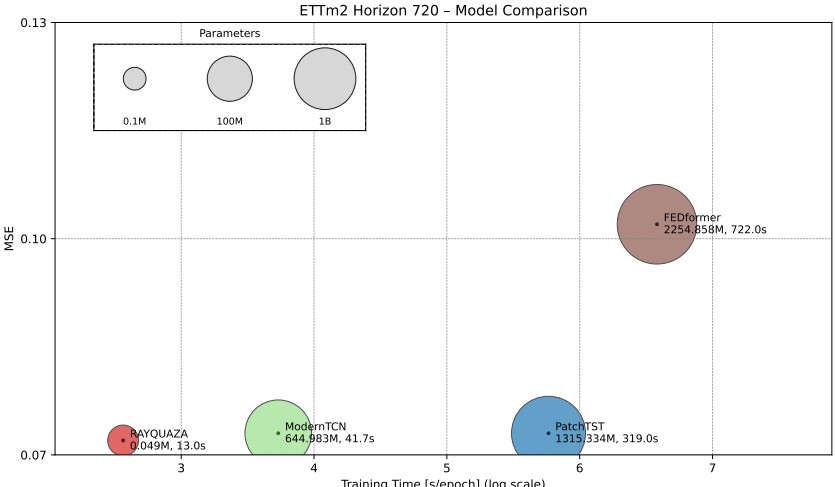

Figure 6: Forecast comparison on the ETTm2 dataset at horizon 720. **RAYQUAZA** closely follows the true target using only 49K parameters, compared to 12.7M in ModernTCN and 666K in TimesNet. The model tracks long-term structure and local variations without overfitting or delay.

**Efficiency on M4.** Table 11 presents a full comparison of **RAYQUAZA**, ModernTCN, and Times-Net in terms of parameter count and per-epoch training time across the four frequency splits of M4. **RAYQUAZA** consistently uses a fraction of the parameters while delivering SOTA accuracy (as shown in Section 5). For instance, on the YEARLY dataset, **RAYQUAZA** requires just 55K parameters more than $200\times$ fewer than ModernTCN (33.9M) and $10\times$ smaller than TimesNet (586K) while reducing per-epoch training time by over 90%.

The trend holds across all other frequency splits: on M4-Quarterly, **RAYQUAZA** uses only 0.09% of ModernTCN's parameters, and just 1.3% compared to TimesNet. Its per-epoch time is 14.07s, compared to 133.42s and 12,191.69s for the transformer-based models. These savings are not only architectural, but computational **RAYQUAZA** dramatically cuts runtime while remaining competitive in performance.

**Efficiency on ETT Benchmarks.** To assess scalability in multistep long-horizon scenarios, we analyze model sizes across four forecast horizons ($T \in \{96, 192, 336, 720\}$) on the ETT benchmarks (ETTh1, ETTh2, ETTm1, ETTm2). Table 14 reports the absolute parameter counts with relative proportions showing how **RAYQUAZA** compares to ModernTCN and TimesNet as a percentage of their parameter counts.

For example, on ETTh1 at horizon 192, **RAYQUAZA** uses only 42K parameters (3.6% of ModernTCN and 7.0% of TimesNet). On ETTm2 at horizon 720, the gap is even more striking: **RAYQUAZA** operates with just 49K parameters, compared to over 12.7M in ModernTCN and 666K in TimesNet. Across all datasets and horizons, **RAYQUAZA**'s relative size never exceeds 11.3% of any baseline model.

**Case Study: ETTm2–720.** Figure 6 illustrates **RAYQUAZA**'s performance on the ETTm2 dataset for horizon 720. Despite using only 49K parameters, less than 1% of the largest baseline **RAYQUAZA** maintains high fidelity to the ground truth, especially in capturing low-frequency oscillations and structural patterns. This reflects the effectiveness of the ATE and iRBF modules in disentangling trend from noise while preserving temporal consistency.

**Summary.** Together, Tables 11 and 14 establish that **RAYQUAZA** offers industry-grade forecasting efficiency: using 1–10% of the parameters and a fraction of the training time required by transformer-based models. This makes it an ideal candidate for deployment in latency-sensitive or compute-constrained environments without compromising accuracy or adaptability.

Table 14: Parameter counts and **RAYQUAZA**'s proportion across horizons on ETT datasets.

| Horizon | Model | ETTh1 | ETTh2 | ETTm1 | ETTm2 |
|---|---|---|---|---|---|
| 96 | RAYQUAZA | 117 552 | 322 421 | 130 703 | 41 783 |
|  | ModernTCN | 659 844 (17.8%) | 659 844 (48.9%) | 2 199 428 (5.9%) | 1 978 244 (2.1%) |
|  | TimesNet | 605 479 (19.4%) | 1 191 943 (27.1%) | 4 708 167 (2.8%) | 1 191 943 (3.5%) |
| 192 | RAYQUAZA | 42 903 | 71 255 | 138 966 (100.0%) | 136 151 |
|  | ModernTCN | 1 176 036 (3.6%) | 1 176 036 (6.1%) | 2 273 252 (6.1%) | 2 715 620 (5.0%) |
|  | TimesNet | 614 791 (7.0%) | 1 201 255 (5.9%) | 4 717 479 (2.9%) | 1 201 255 (11.3%) |
| 336 | RAYQUAZA | 163 643 | 387 385 | 68 849 | 68 849 |
|  | ModernTCN | 1 950 324 (8.4%) | 1 950 324 (19.8%) | 3 489 908 (2.0%) | 3 489 908 (2.0%) |
|  | TimesNet | 628 759 (26.0%) | 1 215 223 (31.9%) | 628 759 (11.0%) | 1 215 223 (5.7%) |
| 720 | RAYQUAZA | 210 407 | 189 849 | 49 124 (100.0%) | 49 124 |
|  | ModernTCN | 4 015 092 (5.2%) | 4 015 092 (4.7%) | 9 425 396 (0.5%) | 12 743 156 (0.4%) |
|  | TimesNet | 666 007 (31.6%) | 1 252 471 (7.4%) | 666 007 (7.4%) | 666 007 (7.4%) |

Our primary claim is **parameter efficiency**, which we define as achieving state-of-the-art accuracy with minimal parameter count. Training time is a secondary context-dependent metric, but we report it for completeness. This section provides a detailed analysis of **RAYQUAZA**'s performance across the full efficiency spectrum.

G.1 POSITIONING ACROSS THE ACCURACY-EFFICIENCY SPECTRUM ON M4

To situate **RAYQUAZA**'s performance, we compare it against models from three distinct tiers on the M4 benchmark: ultra-lightweight linear models, tiny neural networks, and large-scale deep architectures.

As shown in Table 15, **RAYQUAZA** occupies a unique and highly effective position. While it is larger than the ultra-lightweight OLS and FITS models, it delivers a substantial improvement in accuracy, reducing the Weighted SMAPE by over 2.3 points compared to OLS. At the same time, it achieves a better SMAPE score than the much larger ModernTCN while being over **150x smaller**. This confirms

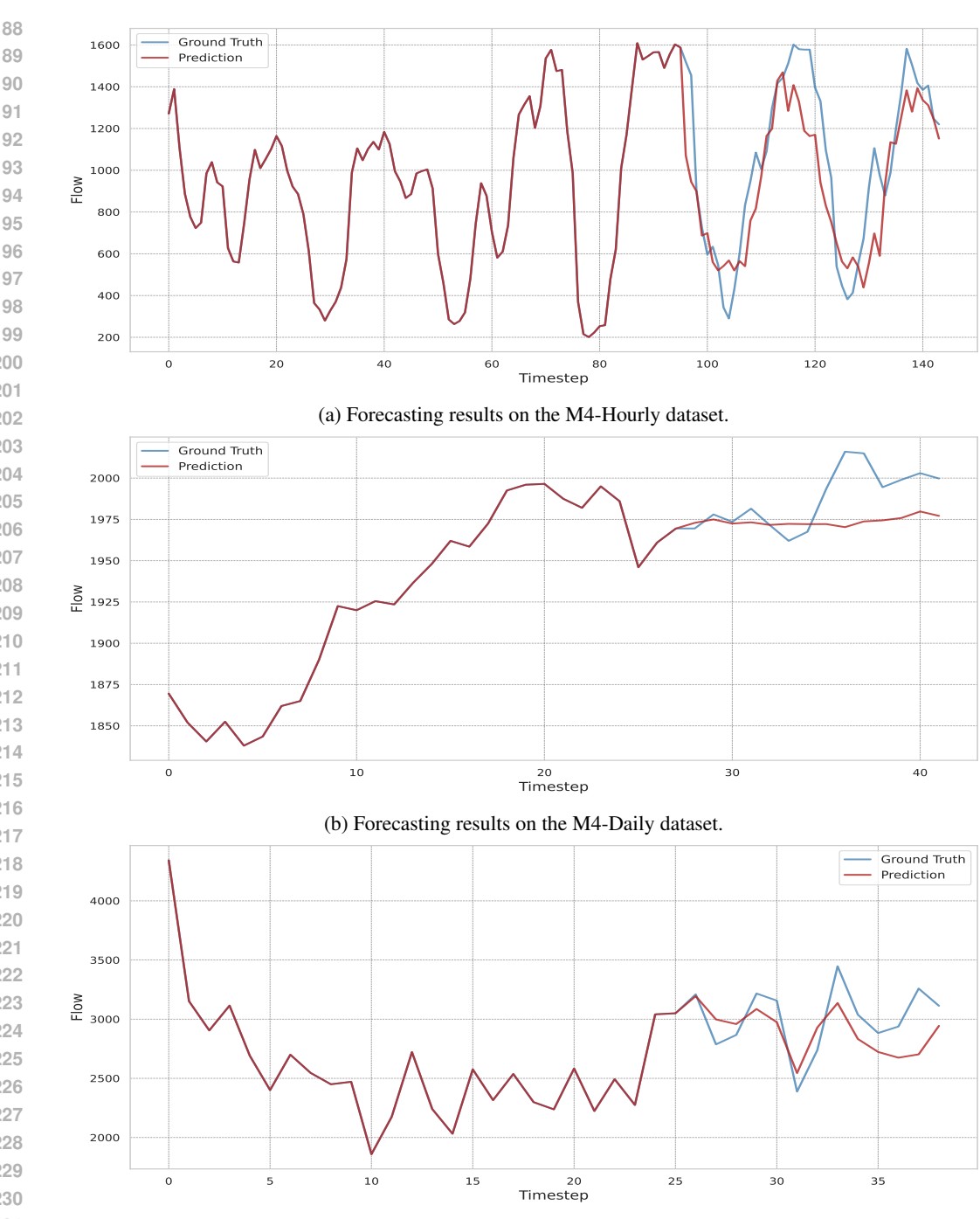

(a) Forecasting results on the M4-Hourly dataset.

(b) Forecasting results on the M4-Daily dataset.

(c) Forecasting results on the M4-Weekly dataset.

Figure 7: Forecasting results using the **RAYQUAZA** model on the M4-Hourly, M4-Daily, and M4-Weekly datasets, demonstrating its performance to follow the overall trend, even if the predictions are not entirely accurate.

**RAYQUAZA**'s ability to bridge the gap, providing state-of-the-art accuracy at a practical, efficient scale.

Table 15: Performance and efficiency comparison across model tiers on the M4 benchmark.

| Model Tier | Representative Model | Parameters | Train Time* | Weighted SMAPE ↓ |
|---|---|---|---|---|
| Ultra-Linear (Speed Anchor) | OLS-full | ≈ 1k | ≈ 1s | 14.02 |
| Tiny NN (Size Anchor) | FITS | 5k–10k | < 60s | — |
| Deep SOTA (Accuracy Anchor) | ModernTCN | 17.4M | ≈ 25 min | 11.70 |
| **Efficient SOTA (Ours)** | **RAYQUAZA** | **80k–120k** | **≈ 4 min** | **11.66** |

*Approximate total training time on M4.

## H  QUALITATIVE ANALYSIS OF THE iRBF REPRESENTATION ON REAL-WORLD DATA

To further validate the adaptive and interpretable nature of the iRBF module, this section provides a deeper qualitative analysis of its learned representations on real-world time series. We analyze the learned basis functions for samples from the M4-Hourly (Figure 9) and M4-Yearly (Figure 8) datasets, which exhibit vastly different temporal structures. These examples demonstrate that the iRBF's adaptive mechanism is not just a theoretical capability but a practical tool that yields meaningful and structured decompositions even in the presence of the noise and non-stationarity characteristic of real-world data.

### H.1  CASE STUDY 1: M4-YEARLY (LOW-FREQUENCY WITH ABRUPT SHOCKS)

The M4-Yearly series follow a slow and smooth trend but include sudden changes that break the pattern. These abrupt events are difficult for classical models to capture. As shown in Figure 8, the iRBF model uses a few wide, low-amplitude basis functions to represent the global trend, while a single narrow and high-amplitude atom isolates the sharp disturbance. This separation keeps the trend clean and avoids distortion. The reconstruction closely matches the original signal (MSE = 0.4852), with the residual shown in panel (a). Panels (b)–(d) display the learned basis functions, their shapes, and amplitudes, while panels (e)–(g) illustrate how the atoms activate over time and combine to form the final signal. Overall, the model adapts efficiently to both stable and irregular structures.

### H.2  CASE STUDY 2: M4-HOURLY (HIGH-FREQUENCY WITH COMPLEX SEASONALITY)

The M4-Hourly dataset contains rapidly changing signals with clear repeating patterns. Figure 9 shows that the iRBF model adopts a different strategy: it learns several narrow atoms that cluster near the origin, forming a shared localized basis. These atoms vary mainly in amplitude, enabling the model to capture fine-scale fluctuations and seasonal patterns. The reconstruction in panel (a) achieves low error (MSE = 0.2246) and follows the fast oscillations of the signal. Panels (b)–(d) show the strongest basis functions, their centers, and amplitude distributions, while panel (e) illustrates how their contributions sum to recreate the full signal. This structure provides flexibility without unnecessary complexity, making it well suited for high-frequency data.

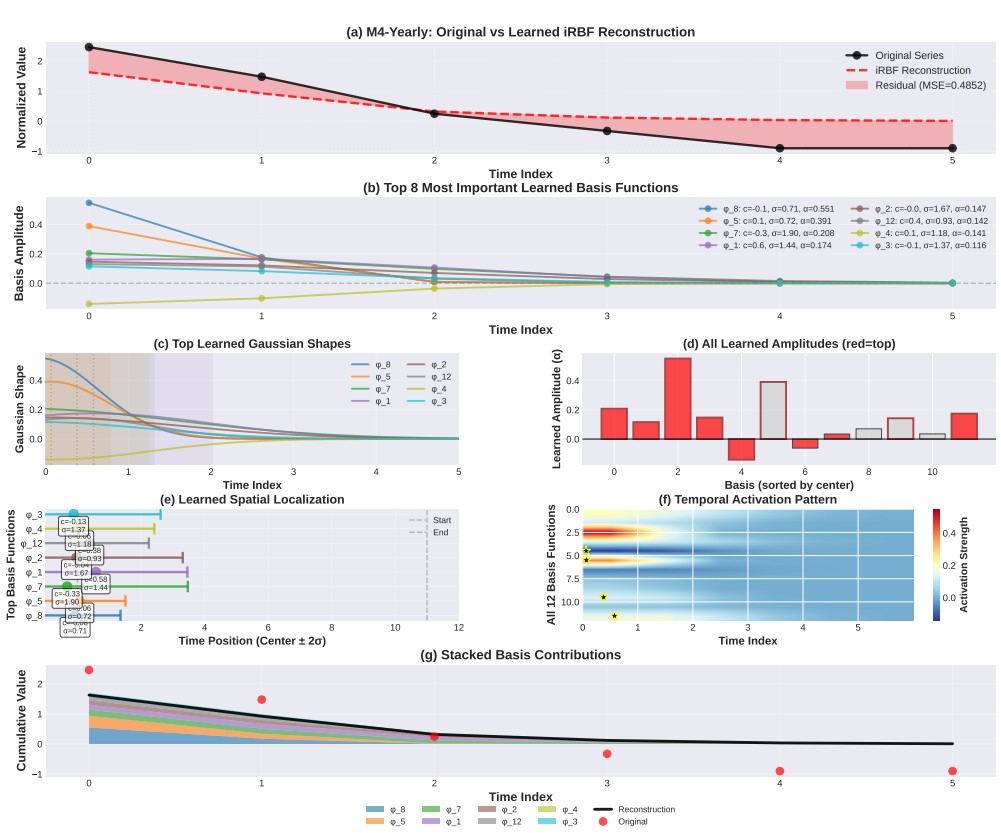

Figure 8: **iRBF Decomposition for the M4-Yearly Series.** (a) Original series, iRBF reconstruction, and residual error (MSE = 0.4852). (b) Eight most influential learned basis functions showing differences in centers, widths, and amplitudes. (c) Gaussian shapes of these top basis functions, distinguishing wide trend atoms from the narrow shock atom. (d) Learned amplitudes for all basis functions, with the dominant atoms highlighted. (e) Spatial localization of the top basis functions, showing centers and effective support. (f) Temporal activation patterns indicating when each basis function contributes. (g) Stacked basis contributions vs. the original signal, showing how atoms combine to reconstruct the series.

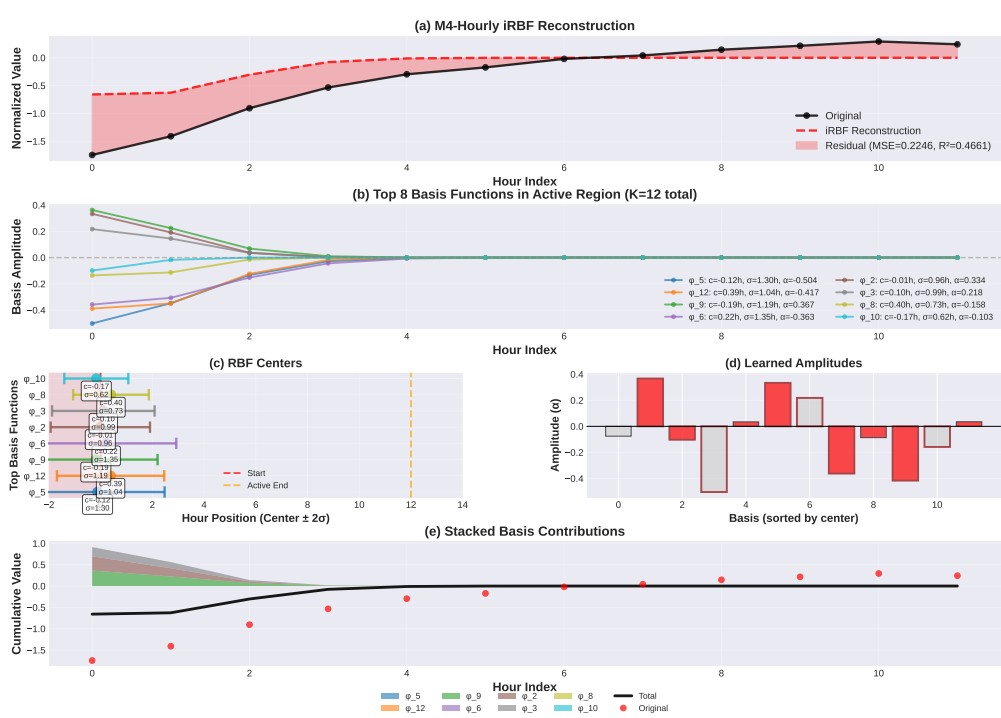

Figure 9: **iRBF Decomposition for the M4-Hourly Series.** (a) Original hourly series, iRBF reconstruction, and residuals (MSE = 0.2246). (b) Eight most influential basis functions in the active region, all narrow and closely centered to capture rapid variations. (c) Distribution of basis centers, showing clustering near the origin. (d) Learned amplitudes for all basis functions, illustrating how the model adjusts atom strengths. (e) Stacked contributions of the basis functions vs. the original signal, showing how localized atoms reconstruct high-frequency and seasonal components.

## I  LIMITATIONS

While **RAYQUAZA** demonstrates state-of-the-art performance and exceptional efficiency, we acknowledge several avenues for future work.

**Advanced Multivariate Modeling.**  Our current work successfully addresses multivariate forecasting through a robust channel-wise (univariate) strategy, where each time-series is modeled independently. This approach is highly effective and parameter-efficient, as demonstrated in our results. However, it does not explicitly model the dynamic cross-channel dependencies that may exist in some complex systems. A primary direction for future work is to extend **RAYQUAZA** to a fully multivariate architecture. This could involve learning a shared basis representation across channels while allowing for channel-specific amplitudes, or integrating cross-channel mixing layers to explicitly capture inter-variable relationships.

**Broader Time-Series Tasks.**  Our evaluation has focused exclusively on forecasting. The adaptive basis representation learned by the iRBF module may also be highly effective for other critical time-series tasks. For example, in anomaly detection, anomalies could be represented by highly localized, high-amplitude Gaussian atoms, providing an interpretable detection mechanism. Exploring the application of our adaptive basis paradigm to tasks like classification and imputation is a promising area for future investigation.

**Theoretical Understanding of the Learned Basis.**  While we provide extensive empirical validation of the learned iRBF representation, a deeper theoretical analysis of its properties would be valuable. Future work could explore the geometric properties of the learned basis space or provide theoretical guarantees on its ability to approximate different function classes of time-series, further solidifying the foundations of the adaptive basis paradigm.

