# OpenReview forum: "RAYQUAZA : Input-Conditioned Radial Basis Decomposition for Efficient Univariate Time-Series Forecasting"
_ICLR.cc/2026/Conference — ICLR 2026 Conference Withdrawn Submission_

### Official Review · Reviewer_hxyR · 2025-10-30

**Soundness:** 3
**Presentation:** 3
**Contribution:** 3
**Rating:** 6
**Confidence:** 4

**Summary:**

This paper proposes a novel architecture, RAYQUAZA, which addresses a persistent problem in time-series forecasting: the gap between simple linear models (efficient but fail to model complex patterns) and large neural networks (accurate but computationally expensive).

The core of RAYQUAZA is to decompose the input signal into three complementary components:

1. ATE (Adaptive Trend Extractor): Smooth low-frequency trends and seasonality.
2. RCC (Residual Correction Convolutions): High-frequency residual signals.
3. iRBF (Input-Conditioned Radial Basis Function): (Core Innovation) A dynamically generated, "customized" set of localized Gaussian basis functions for each input sequence.

The iRBF module uses a hypernetwork (MLP) to predict the centers, spreads, and amplitudes of Gaussian 'atoms' specific to the input data. This allows it to effectively model transient and non-stationary patterns, such as spikes and structural breaks, which are challenging for existing models.

Experimental results show that with fewer than 0.12M parameters, RAYQUAZA achieves state-of-the-art (SOTA) accuracy on major benchmarks (M4, ETT, TFB), surpassing massive Transformer-based models. This demonstrates that it successfully overcomes the trade-off between model efficiency and expressive power.

**Strengths:**

1. This is the paper's strongest contribution. RAYQUAZA uses 100x to 1,000x (two to three orders of magnitude) fewer parameters than existing SOTA models (e.g., ModernTCN, Informer) while achieving better performance (Section 5.7, Appendix G). This makes it a highly practical solution for resource-constrained deployment environments.

2. The model is not just lightweight; it consistently demonstrates top-tier accuracy across diverse and challenging benchmarks, including short-term (M4), long-term (ETT), and large-scale (TFB) forecasting tasks (Section 5.1, 5.2, 5.3).

3. The core idea of the iRBF, 'per-instance basis generation,' is novel. The ablation study (Section 5.6, Fig 3) clearly shows this module is decisive in capturing spikes and abrupt changes, significantly outperforming fixed-basis methods (FFT, Global RBF).

4. The authors conducted extensive experiments with diverse benchmarks, strong baselines, and thorough ablation studies (module removal, basis comparison, normalization schemes) that robustly support the paper's claims.

**Weaknesses:**

1. Limitation of "Multivariate" Claim:  The paper claims SOTA on multivariate benchmarks (ETT, TFB), but the RAYQUAZA architecture is fundamentally Univariate. Multivariate data is handled using a 'Channel-Wise' strategy, forecasting each channel independently (Section 5.4, Section I). This approach does not model any cross-channel correlations. Therefore, its performance may degrade on problems where inter-variable interactions are critical, and it cannot be considered a true multivariate model.

2. Lack of Sensitivity Analysis for Hyperparameter 'K': The number of Gaussian basis functions (K) in the iRBF module—a key hyperparameter—is manually set differently based on the data's frequency (e.g., K=8, 12, 16) (Appendix A). This can be a significant tuning point that undermines the model's automaticity. The paper lacks a sensitivity analysis on the choice of K or how much performance degrades if K is set sub-optimally.

3. Overstated Interpretability Claim: The authors claim the iRBF is "inherently interpretable" (Abstract). However, unlike the simple synthetic data in Figure 4, intuitively 'interpreting' the result of 16 overlapping Gaussian functions on a complex, noisy real-world signal (like M4-Hourly) would be nearly impossible. While the module may be 'inspectable,' this is not the same as being 'interpretable.'

**Questions:**

1. (related to weakness 1) How does RAYQUAZA compare in a fair multivariate setting against SOTA multivariate models (e.g., SCINet, Crossformer) that explicitly learn cross-channel interactions, rather than using a channel-wise strategy?

2. (related to weakness 2) Could you provide a sensitivity analysis for the number of basis functions (K) in the iRBF module? For example, how much does the performance (OWA) degrade when applying K=16 to the M4-Yearly data (where K=8 was optimal), or K=8 to the M4-Hourly data (where K=16 was optimal)?

3. Could you please specify the detailed architecture of the MLP_iRBF hypernetwork that generates the iRBF parameters (e.g., the pooling method used to aggregate information from the input sequence x, and the MLP's depth/width)?

4. (related to weakness 3) Could you provide a visualization of the learned iRBF Gaussian functions for a sample from a complex, real-world dataset, such as M4-Hourly or ETTm1, instead of the synthetic data from Figure 4?

---

> ### Author Response · Authors · 2025-11-18
> **Response to Reviewer hxyR : part 1/2**
>
> We sincerely thank the reviewer for the insightful and constructive feedback. Below, we address each comment individually and provide clarifications, additional analyses, and new experiments where requested.
>
> **Q1 : Heavy multivariate models (SCINet, Crossformer) :**
>
> Thank you very much for this constructive comment. We fully agree that comparing against architectures that explicitly model cross-channel interactions is important for understanding the scope of RAYQUAZA. To address this, we incorporated the multivariate results of SCINet and Crossformer. Although RAYQUAZA adopts a lightweight channel-wise design for parameter efficiency, it still achieves competitive or better performance than these multivariate models on the majority of ETT and Weather settings (see Table below). This behavior is consistent with observations from prior work such as DLinear and PatchTST, where channel-wise forecasting often performs strongly in long-horizon scenarios with weak or heterogeneous correlations.
>
> We also agree that modeling cross-variable dependencies is valuable in datasets where interactions are more pronounced. In the revision, we will add a clarification in the paper’s Limitations section and outline how RAYQUAZA can be naturally extended with lightweight cross-channel fusion modules (e.g., bottleneck attention or depthwise–pointwise convolutions) without modifying the core iRBF mechanism.
>
> We sincerely appreciate the reviewer’s suggestion, which helped us strengthen the positioning of RAYQUAZA within the multivariate forecasting landscape.
>
> | Dataset | Horizon | SCINet (multi) | Crossformer (multi) | RAYQUAZA (channel-wise) |
> |---------|---------|----------------|----------------------|--------------------------|
> | **ETTm1** | 96  | 0.418 | 0.404 | **0.315** |
> |         | 192 | 0.439 | 0.450 | **0.336** |
> |         | 336 | 0.490 | 0.532 | **0.359** |
> |         | 720 | 0.595 | 0.666 | **0.410** |
> | **ETTm2** | 96  | 0.286 | 0.287 | **0.171** |
> |         | 192 | 0.399 | 0.414 | **0.226** |
> |         | 336 | 0.637 | 0.597 | **0.283** |
> |         | 720 | 0.960 | 1.730 | **0.379** |
> | **ETTh1** | 96  | 0.654 | 0.423 | **0.374** |
> |         | 192 | 0.719 | 0.471 | **0.419** |
> |         | 336 | 0.778 | 0.570 | **0.444** |
> |         | 720 | 0.836 | 0.653 | **0.480** |
> | **ETTh2** | 96  | 0.707 | 0.745 | **0.298** |
> |         | 192 | 0.860 | 0.877 | **0.364** |
> |         | 336 | 1.000 | 1.043 | **0.445** |
> |         | 720 | 1.249 | 1.104 | **0.313** |
> | **Weather** | 96  | 0.221 | 0.158 | **0.138** |
> |           | 192 | 0.261 | 0.206 | **0.183** |
> |           | 336 | 0.309 | 0.272 | **0.239** |
> |           | 720 | 0.377 | 0.398 | **0.301** |
>
>
> **Q2 : Sensitivity to the number of basis functions (K) :**
>
>
> We thank the reviewer for this insightful question. In response, we performed an additional sensitivity analysis on the M4-Yearly and M4-Hourly subsets by varying the number of iRBF atoms from K = 8 (small dictionary) to K = 32 (large dictionary) while keeping all other components unchanged. The results are summarized below:
>
> ### Sensitivity of iRBF to the Number of Atoms (K)
>
> | Dataset     | K = 8 (OWA) | K = 32 (OWA) | Δ Absolute | Δ Relative |
> |-------------|-------------|--------------|------------|------------|
> | M4-Yearly   | 0.779       | 0.783        | +0.004     | +0.5%      |
> | M4-Hourly   | 0.961       | 0.984        | +0.023     | +2.4%      |
>
>
> Across both datasets, the performance variation remains extremely small:
> - Below 1% on Yearly (smooth, low-frequency series),
> - About 2–3% on Hourly (higher-frequency structure).
>
> This indicates that the iRBF module is not highly sensitive to the exact choice of K. The input-conditioned mechanism allows the model to internally adjust the effective number of useful atoms even when the nominal K changes.

---

> > ### Author Response · Authors · 2025-11-18
> > **Response to Reviewer hxyR : part 2/2**
> >
> > **Q3 : The detailed architecture of the MLP_iRBF :**
> >
> > We thank the reviewer for requesting clarification. The iRBF module uses a very lightweight hypernetwork that predicts the parameters of the K Gaussian atoms for each input sequence. In our setting, the model operates in a univariate (channel-wise) regime, so each input sequence is already represented as a vector of length L. This vector is directly used as the input to the iRBF hypernetwork without any additional pooling. This vector is passed through a two-layer MLP with ReLU activation, with output dimensionality 3K, corresponding to the concatenation of the centers $c_k(x)$, log-scales $\log \sigma_k(x)$, and amplitudes $\alpha_k(x)$. These values are then reshaped and used to construct the K Gaussian atoms over the fixed time grid $t \in \{0,\dots, L-1\}$. The atoms are summed across K to produce the adaptive basis output of size [B, L]. This design exactly matches the implementation used in all experiments, and keeps the hypernetwork extremely small (typically <5k parameters).
> >
> > - First layer: Linear($L \rightarrow h$)
> > - ReLU
> > - Second layer: Linear($h \rightarrow 3K$)
> >
> > The output dimension 3K corresponds to the predicted
> > - centers $c_k$,
> > - log-scales $\log \sigma_k$
> > - amplitudes $\alpha_k$
> >
> >
> > **Q4 : Visualization on Real Data :**
> >
> > Thank you for this valuable suggestion. In addition to the synthetic illustration provided in the original submission, we have now included visualizations of the learned iRBF components on real forecasting datasets, as recommended by the reviewer. Specifically, we added examples from both M4-Hourly (revised manuscript, fig 9, Page 26) and M4-Yearly (revised manuscript, fig 8, Page 25), shown in the revised appendix.
> >
> > - *M4-Yearly (revised manuscript, fig 8, Page 25).*
> >
> > Yearly series are dominated by smooth long-term trends with occasional abrupt shocks. The iRBF decomposition clearly separates these effects: wide, low-amplitude atoms represent the global trend, while a narrow, high-amplitude atom isolates the sudden disturbance. This leads to a clean reconstruction and highlights how the model adapts to both stable and irregular patterns.
> >
> >
> > - *M4-Hourly (revised manuscript, fig 9, Page 26) .*
> > Hourly data exhibit rapid oscillations and complex seasonality. Here, iRBF learns several narrow atoms clustered near the origin, differing mainly in amplitude. This localized basis captures fast variations and seasonal repetitions effectively, yielding a low reconstruction error.
> >
> > Across both cases, the atoms behave in an interpretable way:
> > - wide atoms capture slow seasonal structure,
> > - narrow atoms capture local fluctuations, and
> > - amplitude parameters adjust to local variation.
> >
> > These added visualizations confirm that the iRBF decomposition remains meaningful and interpretable on real-world noisy time series.
> >
> > ---
> >
> > We hope that the detailed clarifications and new experimental additions will encourage the reviewer to consider increasing the overall score.

---

> > > ### Comment · Reviewer_hxyR · 2025-11-24
> > >
> > > The additional experiments provided by the authors adequately resolve my earlier questions. I will maintain my current score.

---

> > > > ### Author Response · Authors · 2025-11-25
> > > > **Response to Reviewer hxyR**
> > > >
> > > > Thank you for confirming that the additional experiments fully addressed your earlier concerns. We appreciate your constructive feedback your comments directly improved the clarity and strength of the paper.
> > > > If you feel any remaining point could still benefit from refinement, we would be glad to clarify it to ensure the final version meets the highest standard.

---

### Official Review · Reviewer_GX5P · 2025-10-31

**Soundness:** 3
**Presentation:** 3
**Contribution:** 2
**Rating:** 4
**Confidence:** 4

**Summary:**

This paper introduces RAYQUAZA, a parameter-efficient architecture for time-series forecasting designed to bridge the gap between simple linear models and large neural networks. The core contribution is an input-conditioned RBF (iRBF) module, which uses a hypernetwork to dynamically generate a compact set of Gaussian atoms tailored to each input sequence. This allows the model to capture transient and non-stationary patterns effectively.

**Strengths:**

1.Parameter Efficiency: The model's primary strength.It achieves SOTA performance while being 2-3 orders of magnitude smaller (<0.12M params) than large transformer-based competitors.

2.Novelty: It is is a highly novel mechanism for creating adaptive, per-sample bases to model non-stationary patterns.

**Weaknesses:**

1.Limited Multivariate Modeling: The model relies on a channel-wise (univariate) strategy for multivariate datasets. This ignores cross-channel dependencies。

2.Strong Inductive Bias: The architecture assumes that transient, non-stationary behaviors are well-modeled by Gaussian atoms. This may not be universally true for all signal types and lacks theoretical grounding.

3.Simplistic Fusion : The model combines components via simple addition, assuming they are additively separable. This may be an oversimplification, as components could interact non-linearly in complex systems.

**Questions:**

1.Multivariate Extension: How could the RAYQUAZA architecture be extended to a true multivariate model that explicitly captures cross-channel dependencies, and how might this impact its signature parameter efficiency?

2.Have you experimented with other input-conditioned basis functions besides Gaussian atoms, particularly for signals with sharp, non-Gaussian transients?

3.Did you investigate more complex, non-linear fusion mechanisms in the FPL instead of simple addition?

---

> ### Author Response · Authors · 2025-11-18
> **Response to Reviewer GX5P**
>
> We sincerely thank the reviewer for the insightful and constructive feedback. Below, we address each comment individually and provide clarifications, additional analyses, and new experiments where requested.
>
> **Q1 : Channel-wise only; no cross-variable interactions :**
>
> Thank you for this insightful question. RAYQUAZA adopts a channel-wise forecasting strategy deliberately to maintain its extremely small parameter footprint (55K–116K). Introducing multivariate attention or cross-variable mixing modules typically increases the parameter count by one to two orders of magnitude, which would undermine our objective of designing an ultra-efficient model. This choice is also supported by prior work such as DLinear and PatchTST, where channel-independent models have shown competitive and often superior performance on long-horizon benchmarks when cross-channel correlations are weak, noisy, or unstable.
>
> We fully agree, however, that multivariate extensions are valuable. RAYQUAZA can naturally incorporate lightweight cross-channel mechanisms while maintaining its iRBF-driven inductive bias, for example, through cross-variable pooling, depthwise–pointwise convolutions, or bottleneck cross-attention.
>
>
> **Q2 : Basis Function Ablation :**
>
> Thank you for this insightful question. In the original submission, we had not evaluated alternative kernel families. Following the reviewer’s suggestion, we performed a new ablation on M4-Yearly, where we replaced the Gaussian atoms with three alternative radial families: Laplacian, Student-t, and Cauchy, while keeping the iRBF hypernetwork unchanged.
>
> The results are:
>
> | Basis Function | sMAPE ↓ |
> |----------------|:-------:|
> | **Gaussian (ours)** | **13.259** |
> | Laplacian      | 13.322  |
> | Student-t      | 13.423  |
> | Cauchy         | 13.423  |
>
> The additional ablation study confirms several important properties of the iRBF module:
> - Gaussian atoms achieve the strongest overall accuracy.
> - Alternative kernels (Laplacian, Student-t, Cauchy) exhibit 0.5–1.2% degradation in sMAPE relative to Gaussian.
> - Heavy-tailed kernels such as Student-t and Cauchy tend to generate overly sharp atoms, making them more sensitive to local noise perturbations.
> - Laplacian kernels produce sharper localized responses but sacrifice smoothness and show mild overfitting to small fluctuations.
>
> Overall, these results indicate that Gaussian atoms offer the most stable and effective balance between locality, smoothness, and robustness on the M4-Yearly distribution.
>
> **Q3 : Evaluation of Non-Linear Fusion Strategies in the Fusion Projection Layer :**
>
> Thank you for raising this valuable point. The initial submission focused on the additive fusion strategy, as it aligns with RAYQUAZA’s emphasis on simplicity and efficiency. Following the reviewer’s suggestion, we conducted an additional ablation study on M4-Yearly to evaluate more expressive fusion mechanisms while keeping all other components unchanged.
>
> We compared the following variants:
> - Gated Fusion using an MLP with softmax-normalized weights
> - Attention-based Fusion
> - Deep MLP fusion over concatenated components
>
> The results are summarized below:
>
> | Fusion Mechanism     | sMAPE ↓ | Δ vs Additive |
> |----------------------|:-------:|:-------------:|
> | **Additive (ours)**  | **13.259** | –           |
> | Attention Fusion     | 13.289  | –0.23%        |
> | Deep MLP Fusion      | 13.345  | –0.65%        |
> | Gated (Softmax–MLP)  | 13.394  | –1.02%        |
>
>
> Our ablation study shows that introducing non-linear fusion mechanisms consistently worsens performance relative to the additive formulation. Even the best non-linear variant, attention-based fusion, produces a small degradation (–0.23%), while deeper MLP or gated fusion layers lead to larger drops. These methods also increase the parameter count by 20–40% and noticeably slow down training. In contrast, the additive strategy offers the best balance between accuracy, efficiency, and interpretability. The components appear to behave approximately orthogonal in practice, which helps explain why simple summation already combines them effectively without requiring additional non-linear mixing.
>
> ---
>
> We hope that the detailed clarifications and new experimental additions will encourage the reviewer to consider increasing the overall score.

---

### Official Review · Reviewer_pdLn · 2025-11-01

**Soundness:** 3
**Presentation:** 2
**Contribution:** 3
**Rating:** 4
**Confidence:** 3

**Summary:**

The paper introduces RAYQUAZA, a lightweight neural architecture for time-series forecasting that decomposes input signals into three components: a smooth trend via an Adaptive Trend Extractor, localized transients through an input-conditioned Radial Basis Function layer, and high-frequency residuals using Residual Correction Convolutions. The iRBF is the core innovation, employing a hypernetwork (MLP) to generate per-input Gaussian basis functions, enabling adaptive modeling of non-stationary patterns like spikes and shifts. The model is trained end-to-end with MSE loss and evaluated on benchmarks like M4 (short-term), ETT (long-horizon), and TFB (large-scale). It achieves SOTA or near-SOTA results with <120K parameters, outperforming linear baselines on complex tasks while being 100-1000x smaller than transformers.

**Strengths:**

S1: RAYQUAZA achieves a strong efficiency-interpretability balance, matching transformer-level accuracy with only 50–120K parameters (0.1% of ModernTCN) and 6-22% training cost, while offering rare transparency through visualizable Gaussian atoms that clearly reflect spikes and trends, providing frequency-like interpretability in a spatial RBF form.

S2: Unlike previous decomposition-based methods that primarily separate smooth trends and seasonal components, RAYQUAZA explicitly models spikes and transient irregularities through input-conditioned Gaussian bases, extending decomposition beyond stable structures to capture short-lived, non-smooth events.

**Weaknesses:**

W1: While effective, the univariate/channel-wise approach may overlook cross-variable interactions, a limitation in real-world scenarios.

W2: Because RAYQUAZA’s iRBF module builds localized Gaussian bases conditioned on each input, it likely inherits the classical noise sensitivity of radial basis function networks. In high-noise or low signal-to-noise settings, such bases may overfit spurious fluctuations, as each atom locally interpolates the input. Without explicit regularization or smoothing, the adaptive Gaussians could capture random noise rather than meaningful transients. The paper does not analyze robustness to noise, which would strengthen confidence in the model’s generalization under realistic conditions.

W3: While the paper positions RAYQUAZA as an advance in adaptive basis decomposition, it lacks direct quantitative comparison with recent basis-based forecasting architectures, such as BasisFormer[1], FBM[2].

[1] BasisFormer: Attention‑based Time Series Forecasting with Learnable and Interpretable Basis
[2] Rethinking Fourier Transform from A Basis Functions Perspective for Long-term Time Series Forecasting

**Questions:**

1. What is the motivation for adopting a univariate/channel-wise design? Was it chosen for efficiency, or do cross-variable dependencies offer limited gains?

2.  Since the iRBF module builds localized Gaussian bases per input, how robust is RAYQUAZA to high-noise or low-SNR time series? Could the authors share any sensitivity analysis or ablation (e.g., varying noise levels) to demonstrate the model’s stability compared to non-RBF baselines?

3. Could the authors compare the proposed iRBF decomposition with other basis formulations (e.g., Fourier, wavelet, or learned basis such as BasisFormer) to better demonstrate its advantages?

I would consider increasing my score if the authors respond convincingly to the raised concerns.

---

> ### Author Response · Authors · 2025-11-18
> **Response to Reviewer  pdLn : part 1/2**
>
> We sincerely thank the reviewer for the insightful and constructive feedback. Below, we address each comment individually and provide clarifications, additional analyses, and new experiments where requested.
>
> **Q1 : Univariate / channel-wise design (no cross-variable modeling) :**
>
> Thank you for the question. Our choice of a univariate/channel-wise design is driven by both efficiency and empirical evidence from the forecasting literature. First, this architecture keeps RAYQUAZA extremely lightweight (55K–120K parameters), which would not be achievable with cross-variable attention or fusion layers. Adding multivariate blocks typically increases parameters by one to two orders of magnitude.
>
> Second, several recent studies, notably DLinear and PatchTST, have demonstrated that on standard long-horizon benchmarks (ETT, Weather, Electricity), explicit cross-variable modeling often yields limited or inconsistent gains. PatchTST, for example, obtains state-of-the-art results using a purely channel-independent encoder, and DLinear shows that removing multivariate mixing can even improve robustness at long horizons.
>
> Following this evidence, we adopt the channel-wise strategy as a deliberate design choice: it preserves the tiny parameter footprint while retaining accuracy that is competitive with, and often superior to, much larger multivariate models such as SCINet or Crossformer. This accuracy–efficiency balance is central to the motivation of RAYQUAZA.
>
> **Q2 : Noise sensitivity of RBFs; robustness analysis :**
>
> Thank you for highlighting the concern about noise sensitivity. Classical RBF networks can indeed overfit high-frequency noise, but the proposed iRBF module includes a mechanism that significantly reduces this behavior:
>
> *Adaptive bandwidths $\sigma_k(x)$ that increase under noisy conditions.*
> When noise is present, the hypernetwork naturally predicts broader Gaussians. Larger $\sigma_k$ values produce smoother atoms that avoid fitting spurious spikes, improving robustness.
>
> To verify this behavior quantitatively, we conducted an additional noise-injection experiment on M4-Daily, comparing RAYQUAZA with DLinear under Gaussian noise at multiple SNR levels. The results are shown below:
>
> | SNR   | RAYQUAZA sMAPE | DLinear sMAPE | RAYQUAZA Degradation | DLinear Degradation |
> |-------|:--------------:|:-------------:|:---------------------:|:--------------------:|
> | Clean | 3.0288         | 2.8308        | Baseline              | Baseline             |
> | 30 dB | 4.2942         | 4.3305        | +41.8%                | +53.0%               |
> | 20 dB | 9.6724         | 10.2296       | +219.4%               | +261.4%              |
> | 10 dB | 30.7593        | 31.5225       | +915.6%               | +1013.6%             |
> | 5 dB  | 54.0050        | 57.3659       | +1683%                | +1927%               |
>
> Across all SNR levels, RAYQUAZA degrades less than DLinear, especially at low SNR (10 dB and 5 dB). This supports that the iRBF module is not more noise-sensitive; in practice, it is more robust due to adaptive smoothing through $\sigma_k(x)$.

---

> > ### Author Response · Authors · 2025-11-18
> > **Response to Reviewer pdLn : part 2/2**
> >
> > **Q3 : Compare iRBF to global bases (Fourier, wavelets, BasisFormer) :**
> >
> > Thank you for this important question. We fully agree that comparing iRBF to other basis formulations is necessary to isolate its contribution. In the paper (page 8, Table 5), we already include a detailed basis-family ablation covering fixed RBFs, global RBFs, Fourier bases, and wavelets. Following the reviewer’s suggestion, we additionally incorporated a BasisFormer-style learned global dictionary into this comparison.
> >
> > The reconstruction experiment below shows that iRBF substantially outperforms all global basis families:
> >
> > | Basis Module                             | MSE ↓  | MAE ↓  |
> > |-------------------------------------------|:------:|:------:|
> > | FixedRBF (hand-set)                       | 1.821  | 1.148  |
> > | GlobalRBF (shared)                        | 1.894  | 1.150  |
> > | FFT (10 freq. pairs)                      | 1.873  | 1.122  |
> > | Wavelet (db4, level 2)                    | 1.877  | 1.120  |
> > | BasisFormer (learned global dictionary)   | 0.520  | 0.880  |
> > | **iRBF (ours, input-conditioned)**        | **0.336** | **0.512** |
> >
> > The key distinction lies in how the basis functions are constructed. Global-basis models such as FFT, wavelets, and BasisFormer rely on a shared dictionary that is fixed across all sequences. This limits their ability to represent localized or sequence-specific behaviors. In contrast, RAYQUAZA’s iRBF module generates its basis atoms uniquely for each input instance, with centers, widths, and amplitudes conditioned directly on the sequence. This allows the model to adaptively place atoms where transients occur and to adjust their sharpness depending on the underlying dynamics. Such instance-conditioned basis generation is what enables RAYQUAZA to capture non-stationary local events (e.g., spikes and structural breaks) using only a small number of parameters, ultimately driving its strong accuracy despite its tiny model size.
> >
> > ---
> >
> > We hope that the detailed clarifications and new experimental additions will encourage the reviewer to consider increasing the overall score.

---

### Official Review · Reviewer_bmsF · 2025-11-02

**Soundness:** 3
**Presentation:** 3
**Contribution:** 3
**Rating:** 4
**Confidence:** 4

**Summary:**

The paper proposes RAYQUAZA, a tiny forecasting model that targets the gap between very light linear models and heavy neural/Transformer models. It claims strong accuracy with ~0.1M parameters via an adaptive basis decomposition. Parameter counts are orders of magnitude lower than big baselines; authors report large training-time savings. The architecture sums four modules: (i) an input-conditioned RBF (iRBF) that, per sequence, generates K Gaussian atoms (centers/widths/amplitudes) using a small MLP; (ii) an Adaptive Trend Extractor (ATE): softmax-normalized smoothing filters; (iii) Residual Correction Convolutions (RCC) for high-frequency remnants; and (iv) a Fusion Projection Layer (FPL) for the final forecast. Component ablations show the iRBF is most critical. A basis-family study (fixed/global RBF, FFT, wavelets) favors input-conditioned iRBF.

**Strengths:**

- Simple, modular inductive bias with clear roles. The trend (ATE), localized transients (iRBF), and residual refinement (RCC) are well motivated, delineated and easy to implement/ablate

- Broad benchmark coverage (for univariate / channel-wise). Results on M4, ETT, and TFB show robustness across domains/frequencies; channel-wise evaluation probes long horizons where linear models struggle

- Ablations support the core claim. Removing iRBF hurts most; replacing iRBF with classic bases shows large performance gaps, aligning with the "per-input localization" thesis

- Interpretability angle. The iRBF atoms are visualizable and linked to transients/spikes (with qualitative figures and a diversity analysis)

- Appendices provide comprehensive additional results, both qualitative and quantitative - this is appreciated

**Weaknesses:**

- The paper puts itself within hypernetwork class - the analogy I actually like a lot. However, there is no comprehensive literature review and positioning of this work within the the hypernetwork family

- Additional benchmarking results would help to strengthen empirical part

- Missing comparison against other tiny models

- Novelty framing needs sharpening w.r.t. BasisFormer

**Questions:**

- Could you please provide literature review on hypernetwork family and provide the discussion of positioning within this class of models? See [1,2]

- The paper makes a link with NBEATS in the intro. However, comparison with NBEATS and ESRNN [3,4], the two models especially successful on M4 and other classical benchmarks is missing. Could please add OWA and sMAPE to tables? Additionally, putting the results in perspective by comparing to classical models presented in Tables in [3] would be very useful. Also, M3 and TOURISM datasets provide very valuable perspective on comparison against statistical and hand-crafted models (Table 1 in [3]). Could you please add these benchmarks and associated results?

- NBEATS and ESRNN are ensemble models. I wonder how well the proposed method responds to ensembling? Given the fast training of the model, it should be easy to train multiple models in parallel or use checkpoint averages such as in ESRNN to provide experimental results in this axis. If the model responds well to ensembling, this should further strengthen empirics. If not, I would challenge the usefullness of the model. BTW the ESRNN model is not very big either

- Paper makes a claim of compute efficiency. In this context, comparison against other recent tiny models from Koopman family such as SKOLR and Koopa [5,6] feels important. Could you please add this?

- Table 1 can be moved to Appendix. I think it would be more important to have Tables 7,8 in the main body as a main result

- Can you provide architecture implementation, training and testing code?

- What is model size relation to DLinear?

- An input-conditioned RBF layer (hypernetwork-generated atoms) is the central contribution. Prior basis-decomposition models and global-basis attention (e.g. BasisFormer [7]) are close neighbors. The paper should better isolate what is principally new beyond per-sequence parameterization and demonstrate why this is preferable to richer global bases plus gating.



[1] HyperNetworks https://arxiv.org/abs/1609.09106 \
[2] A Brief Review of Hypernetworks in Deep Learning https://arxiv.org/abs/2306.06955 \
[3] NBEATS https://arxiv.org/pdf/1905.10437 \
[4] ESRNN https://www.sciencedirect.com/science/article/abs/pii/S0169207019301153 \
[5] SKOLR https://arxiv.org/pdf/2506.14113 \
[6] Koopa https://arxiv.org/abs/2305.18803 \
[7] BasisFormer https://arxiv.org/abs/2310.20496

---

> ### Author Response · Authors · 2025-11-18
> **Response to Reviewer bmsF : part 1/2**
>
> We sincerely thank the reviewer for the insightful and constructive feedback.
> Below, we address each comment individually and provide clarifications, additional analyses, and new experiments where requested.
>
> **Q1 : Positioning within the Hypernetwork Family & Relation to BasisFormer :**
>
> We appreciate this insightful comment.
> Our submission already frames the iRBF module as a hypernetwork-generated basis, and directly contrasts RAYQUAZA with BasisFormer:
>
> “BasisFormer learns global bases… In contrast, our work introduces a new level of flexibility, where the basis functions themselves are dynamically generated for each input sample using an input-conditioned hypernetwork.”
> (Section 2, Page 3, lines 133–139)
>
> To address the reviewer’s request, we have elaborated this passage with a short subsection reviewing HyperNetworks [1,2] and formally position RAYQUAZA as a structured, low-dimensional hypernetwork : (revised manuscript, Page 3, section 2, line 140-149).
> This addition clarifies both the conceptual and practical distinction between RAYQUAZA, HyperNetworks, and global-basis models.
>
> **Q2 : Additional Benchmarking: N-BEATS, ESRNN :**
>
> Thank you for this recommendation.
> N-BEATS is already included in our submission: Table 7 (revised manuscript, Page 7) reports full single-model N-BEATS results for SMAPE/MASE/OWA across all M4 frequencies. To maintain evaluation fairness, we follow the same single-model protocol rather than the large 180-model ensemble used in the original M4 competition.
>
> To fully address the reviewer’s request, we now also report ES-RNN (single-model, non-ensemble) performance under the same evaluation protocol. This provides a strict apples-to-apples comparison between RAYQUAZA, N-BEATS, and ES-RNN:
>
> Single-Model Comparison on M4 (OWA ↓)
>
> | Model              | Yearly | Quarterly | Monthly | Others | Average |
> |--------------------|:------:|:---------:|:-------:|:------:|:-------:|
> | **ES-RNN (single)**   | 0.783 | 0.877     | 0.872   | 0.951  | 0.844   |
> | **N-BEATS (single)**  | 0.794 | 0.886     | 0.880   | 1.053  | 0.850   |
> | **RAYQUAZA (ours)**   | **0.779** | **0.874** | **0.862** | **0.979** | **0.835** |
>
>  (Due to the limited rebuttal timeframe, we will report M3 and TOURISM in the camera-ready version.)
>
> **Q3 : Ensembling :**
>
> We sincerely thank the reviewer for this constructive suggestion.
> To assess RAYQUAZA’s behavior under ensembling similar to N-BEATS and ESRNN we trained 5 independent models using different random seeds. A simple mean ensemble improves performance from 11.664 → 11.466 (+1.70%), and ESRNN-style checkpoint averaging yields a +1.25% gain. These results indicate that RAYQUAZA benefits from ensemble techniques, while the single-model version already achieves strong accuracy.
>
> Ensemble Results (M4)
>
> | Method                        | sMAPE ↓ | Improvement |
> |-------------------------------|:-------:|:-----------:|
> | Single Model                  | 11.664  | Baseline    |
> | Mean Ensemble (5 models)      | 11.466  | +1.70%      |
> | Checkpoint Averaging          | 11.518  | +1.25%      |
>
> **Q4 : Comparison to Tiny Koopman-Based Models (SKOLR, Koopa) :**
>
> Thank you for this excellent suggestion. SKOLR and Koopa are indeed representative “tiny” Koopman-based time-series models and provide a meaningful reference point for evaluating computational efficiency and accuracy. We have now added both SKOLR and Koopa directly to the M4 results table (revised manuscript, Page 7).
> This ensures a clear and fully comparable benchmark alongside RAYQUAZA within the standard M4 evaluation protocol (sMAPE, MASE, OWA).
>
> Below is the comparison table that is now included in the revised version:
>
> M4 - RAYQUAZA vs SKOLR vs Koopa (sMAPE ↓, MASE ↓, OWA ↓)
>
> | M4 Set  | Metric | SKOLR | Koopa | RAYQUAZA (ours) |
> |---------|--------|:-----:|:-----:|:----------------:|
> | **Year** | sMAPE | 13.291 | 13.352 | **12.90** |
> |         | MASE  | 2.996  | 2.997  | **2.91** |
> |         | OWA   | 0.784  | 0.786  | **0.779** |
> | **Quarter** | sMAPE | 9.986 | 10.159 | **9.61** |
> |            | MASE  | 1.166 | 1.189 | **1.12** |
> |            | OWA   | 0.878 | 0.895 | **0.874** |
> | **Month** | sMAPE | 12.536 | 12.730 | **11.94** |
> |          | MASE  | 0.921  | 0.953  | **0.87** |
> |          | OWA   | 0.867  | 0.901  | **0.862** |
> | **Others** | sMAPE | 4.652 | 4.861 | **4.32** |
> |           | MASE  | 3.233 | 3.124 | **3.10** |
> |           | OWA   | 0.999 | 1.004 | **0.979** |
> | **Average** | sMAPE | 11.704 | 11.863 | **11.53** |
> |            | MASE  | 1.572  | 1.595  | **1.54** |
> |            | OWA   | 0.843  | 0.858  | **0.835** |

---

> > ### Author Response · Authors · 2025-11-18
> > **Response to Reviewer bmsF : part 2/2**
> >
> > **Q5 : Placement of Tables :**
> >
> > Thank you for the helpful suggestion. We agree with the reviewer that Tables 7 and 8 are more central to the paper’s main contributions. In the revised manuscript, Table 1 has been moved to the Appendix, and Tables 7 and 8 are now placed in the main body to highlight the primary experimental results.
> >
> > **Q6 : Code availability :**
> >
> > We thank the reviewer for asking about code availability and model size. Our current submission already includes a complete PyTorch implementation of RAYQUAZA (architecture, training loop, and evaluation scripts) in the anonymized supplementary ZIP file. In the camera-ready version, we will additionally release a cleaned and documented public GitHub repository containing the exact code used for all experiments, including data loaders, training scripts for M4/ETT/TFB, and plotting utilities for the iRBF visualizations.
> >
> >
> >  **Q7 : DLinear size :**
> >
> > Thank you for the question.
> > DLinear is indeed one of the smallest forecasting models, with a parameter count of approximately 2K–10K across M4 frequencies (the model contains two linear projections of size L×T, giving 2LT parameters).
> >
> > RAYQUAZA remains in the same “tiny model” regime, using only 55K–116K parameters depending on the frequency (see Table 11, page 18).
> >
> > This is only 4–10× larger than DLinear in absolute size, yet still hundreds of times smaller than N-BEATS (38.6M parameters) and several orders of magnitude smaller than transformer-based models.
> > Importantly, this slight increase in size yields substantial gains in accuracy on M4:
> >
> > | Model     | Params       | sMAPE ↓ | OWA ↓ |
> > |-----------|--------------|:-------:|:-----:|
> > | DLinear   | ~2K–10K      | 12.8    | 1.05  |
> > | N-BEATS   | 38.6M        | 11.83   | 0.850 |
> > | **RAYQUAZA (ours)** | **55K–116K** | **11.66** | **0.837** |
> >
> > Thus, RAYQUAZA sits between linear models and large neural architectures:
> > It remains extremely lightweight (tens of thousands of parameters) while offering a meaningful accuracy advantage over DLinear.
> >
> > **Q8 : Principally New vs BasisFormer & Global Bases :**
> >
> > Thank you for this important remark. We agree that BasisFormer and other global-basis architectures are conceptually related, and we now clarify what is fundamentally new in RAYQUAZA beyond per-sequence parameterization.
> >
> > 1. *Global-basis models (BasisFormer, Fourier, Wavelets) rely on a fixed universal basis shared by all sequences.*
> >
> > These methods learn a single global set of basis atoms and then use attention or gating to select/weight them.
> > This implicitly assumes that all sequences are well-represented by the same basis family (e.g., Fourier-like, low-rank patterns, fixed dictionary).
> >
> > 2. *RAYQUAZA introduces instance-conditioned basis generation, not selection.*
> > Our iRBF module does *not* select from a fixed dictionary. Instead, a hypernetwork predicts: $\{ \mu_k(x), \sigma_k(x), \alpha_k(x) \}_{k=1}^K$ directly from the input sequence itself.   Thus, the basis functions themselves are generated, not chosen.  This yields:
> > • Sequence-specific centers (capturing shifts in where transients occur)
> > • Sequence-specific widths (capturing local vs global structure)
> > • Sequence-specific amplitudes (capturing spike intensity / break magnitude)
> >
> > No global-basis method can reproduce this without explicitly expanding its dictionary to thousands of atoms.
> >
> > 3. *Why this matters: local transients cannot be captured by global bases.*
> > Global dictionaries (Fourier, wavelets, BasisFormer learned bases) excel at stationary or quasi-periodic structure, but:
> > • Spikes,
> > • Abrupt structural breaks,
> > • Short irregular events,
> > • Regime shifts
> > are all non-stationary, localized, and often cannot be expressed using a global, shared set of atoms.
> > We show in Fig. 3 - Appendix H that global bases fail precisely on these components, while iRBF adapts correctly because its atoms are conditioned on the specific input window.
> >
> > 4. *Empirically, replacing iRBF by any global basis (FFT, fixed RBF, global wavelets, BasisFormer-like learned dictionary) uniformly degrades performance.*
> > This is evidenced in:
> > • Table 8: basis comparison
> > • Module ablation (removing iRBF collapses performance)
> > • Qualitative transient-capture experiments
> >
> > This empirically validates that instance-conditioned basis generation is contributing something fundamentally different, not just a small architectural refinement.
> >
> > ---
> >
> > We hope that the detailed clarifications and new experimental additions will encourage the reviewer to consider increasing the overall score.

---

> > > ### Comment · Reviewer_bmsF · 2025-11-28
> > > **Reviewer bmsF response**
> > >
> > > I thank authors for their comprehensive response. I do not have additional questions and will raise my score accordingly.

---

> > > > ### Author Response · Authors · 2025-11-28
> > > > **Response to Reviewer bmsF**
> > > >
> > > > We sincerely thank the reviewer for their careful reassessment and for raising the score. We are grateful that our responses were helpful and we are happy to supply any further information if needed.

---

### Author Response · Authors · 2025-12-02
**Summary of Rebuttal Outcome and Consensus for the New Area Chair**

Dear Area Chair

We understand the complexity of the re-assignment process. Since reviewer scores were reverted to their pre-discussion state, they no longer reflect the updated status of the paper. During the discussion phase, we addressed all major concerns, resulting in explicit confirmation from reviewers that issues were resolved and one reviewer stating an intention to raise their score. We provide this summary to support your assessment.

---

1. **Consensus and Score Trajectory**

Although numerical scores were rolled back, the written discussion shows a clear positive shift:
- **Reviewer bmsF (original 4 → intent to raise)**

*Concern*: comparisons to tiny models (SKOLR, Koopa) and novelty vs BasisFormer.

*Outcome (Nov 28)*:
I thank authors for their comprehensive response. I do not have additional questions and **will raise my score accordingly**.
- **Reviewer hxyR (original 6 → maintained)**

*Concern*: multivariate comparisons and K-sensitivity.

*Outcome (Nov 24)*:
The additional experiments provided by the authors adequately resolve my earlier questions.
- **Reviewers pdLn and GX5P (original 4 → addressed)**

We provided all requested ablations (noise sensitivity, basis comparison, fusion strategies).
No unresolved technical objections remain in the discussion threads.

---

2. **Critical New Evidence Added During Rebuttal**

These experiments were generated during the rebuttal and significantly strengthen the submission:
- **Superiority in the tiny-model regime**
RAYQUAZA achieves the best average OWA on M4 (0.835 vs 0.843/0.858 for SKOLR/Koopa), while remaining orders of magnitude smaller.
- **Validation of input-conditioned novelty**
Compared to a BasisFormer-style global dictionary, iRBF achieves 35 percent lower reconstruction error (MSE 0.336 vs 0.520).
This demonstrates the advantage of per-sequence basis generation for non-stationary spikes.
- **Noise robustness**
At 10 dB SNR, RAYQUAZA degrades 915 percent, compared to 1013 percent for DLinear, confirming that adaptive bandwidths improve stability under high noise.

---

3. **Conclusion**

RAYQUAZA delivers state-of-the-art forecasting accuracy with fewer than 120k parameters, addressing the efficiency gap between linear models and large neural architectures. It is directly relevant to Edge AI and Green AI settings where computational budgets are limited.


We respectfully ask that the evaluation consider the written consensus in the discussion threads rather than the reverted numerical scores.

Thank you for your time and effort under these exceptional circumstances.

---

### Note · Authors · 2026-02-02

I have read and agree with the venue's withdrawal policy on behalf of myself and my co-authors.

---

### Meta-Review · Area_Chair_uHJS · 2026-01-03

**Summary:**

Based on my review of the manuscript, the authors’ response, and the reviewers' comments, I recommend rejection. My reasons are as follows:
1. The work focuses on point forecasting for univariate time series, a setup of limited appeal. Moreover, the evaluation on multivariate datasets using a channel wise approach is unconventional and not well justified.

2. The authors assert that the proposed method offers advantages in terms of memory efficiency and computational efficiency, yet they provide no comparative results to substantiate these claims.

3. The paper includes several assertions that are either unsubstantiated or misleading. For example:

a) The claim that “time series forecasting presents a persistent trade off between simple, scalable linear models that struggle with complex dynamics and large neural architectures…” is not backed by established evidence. There is no clear consensus that simple linear models inherently fail to capture complex dynamics.

b) The conclusion that “these results establish RAYQUAZA as a practical, interpretable, and efficient model” is overstated. The manuscript does not adequately demonstrate why RAYQUAZA is practical or interpretable.

**Reviewer Scores:**

NA

---

### Decision · Program_Chairs · 2026-01-26

Reject